# Concerto: Joint 2D-3D Self-Supervised Learning Emerges Spatial Representations

Yujia Zhang[1]    Xiaoyang Wu[1]    Yixing Lao[1]    Chengyao Wang[2]
Zhuotao Tian[3]    Naiyan Wang    Hengshuang Zhao[1]*
[1]The University of Hong Kong    [2]The Chinese University of Hong Kong
[3]Harbin Institute of Technology (Shenzhen)
https://pointcept.github.io/Concerto

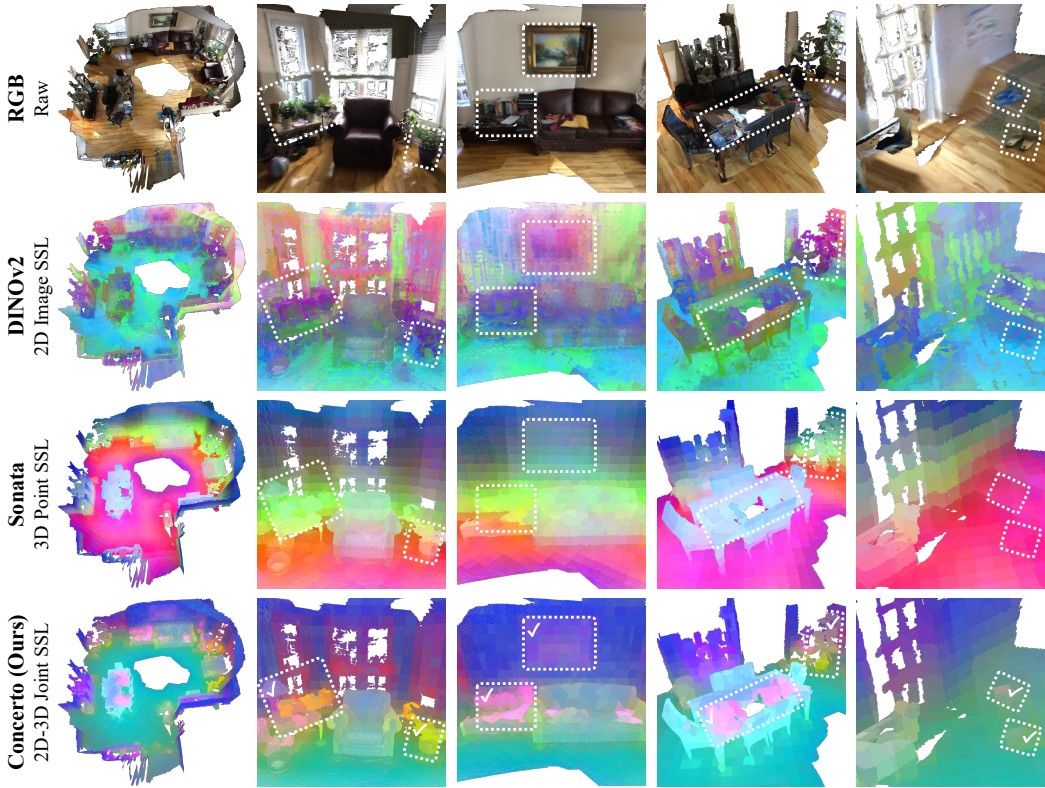

Figure 1: We visualize the principal components of point features learned by 2D and 3D self-supervised models [28, 49], mapped to RGB colors. DINOv2 lacks geometric awareness, and Sonata struggles to capture fine textures. Concerto integrates intra-modal self-distillation with cross-modal joint embedding prediction, enabling a self-supervised point cloud transformer [47] to learn richer, emerging spatial representations with fine-grained geometric and semantic consistency across views.

## Abstract

Humans learn abstract concepts through multisensory synergy, and once formed, such representations can often be recalled from a single modality. Inspired by this principle, we introduce Concerto, a minimalist simulation of human concept learning for spatial cognition, combining 3D intra-modal self-distillation with 2D-3D cross-modal joint embedding. Despite its simplicity, Concerto learns more coherent and informative spatial features, as demonstrated by zero-shot visualizations. It outperforms both standalone SOTA 2D and 3D self-supervised models by 14.2% and 4.8%, respectively, as well as their feature concatenation, in linear probing for 3D scene perception. With full fine-tuning, Concerto sets new SOTA results across multiple scene understanding benchmarks (e.g., 80.7% mIoU on

---

*Corresponding author

39th Conference on Neural Information Processing Systems (NeurIPS 2025).

ScanNet). We further present a variant of Concerto tailored for video-lifted point cloud spatial understanding, and a translator that linearly projects Concerto representations into CLIPs language space, enabling open-world perception. These results highlight that Concerto emerges spatial representations with superior fine-grained geometric and semantic consistency.

# 1  Introduction

Learning strong spatial representations in a self-supervised manner is foundational for spatial cognition tasks, spanning from low-level machine perception to high-level reasoning in domains such as autonomous driving [8, 38], mixed reality [7, 23, 37], and robotics [17]. Recent advances in self-supervised learning have significantly improved foundational representation models for the two dominant spatial modalities: 2D images [4, 12, 18, 51, 56] and 3D point clouds [29, 46, 49, 50, 53]. Without the need for human annotations, these models have demonstrated strong performance across various downstream tasks by enabling the learning of geometry, and semantics at scale.

However, despite their individual successes within each data modality, our pilot study reveals that self-supervised representations learned independently from images and point clouds do not fully overlap. Specifically, concatenating features from self-supervised image models (e.g., DINOv2 [28]) and point cloud models (e.g., Sonata [49]) leads to improved linear probing performance, suggesting that each modality captures complementary, rather than redundant, aspects of spatial information. The observation hints at the existence of *a more robust and rich feature space that emerges from the interaction between 2D and 3D modalities*, indicating the core aim of this research: *to uncover superior spatial representations through multi-modal self-supervised learning*.

Our inspiration toward this target is rooted in how humans learn abstract concepts: through multisensory synergy [6, 36]. Consider the example of an *apple* (as illustrated in Fig. 2)our understanding of it is not limited to its visual appearance. Instead, the concept is formed through repeatedly seeing, touching, and tasting apples, allowing us to internalize its geometry, texture, and semantic meaning in a unified, predictive way. This cognitive process reflects a continuous, cross-modal integration of sensory data into a grounded concept space (Fig. 2 top). Yet once such a representation is formed, it can be evoked from just a single modality: seeing an image of an apple can vividly recall its weight and texture, just as holding one can bring its color and shape to mind (Fig. 2 bottom). This ability to retrieve rich, structured knowledge from partial sensory input underscores the importance of learning modality-agnostic representations that are both unified and predictive.

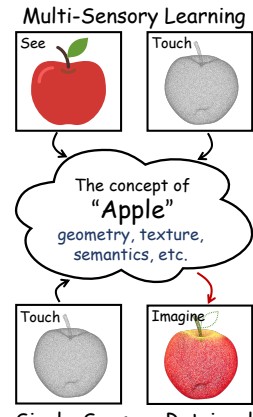

Figure 2: **The "Apple" concept in cognition.**

Driven by this vision, our methodology aims to offer a simple yet effective imitation of human multisensory synergy. To this end, we compose a **Concerto** of 2D-3D joint self-supervised learning, coupling intra-modal self-distillation for point representations [28, 49] with cross-modal joint embedding prediction from images to point clouds [26]. Ultimately, this training yields a self-supervised PTv3 [47] model, pretrained on 40k raw point clouds and 300k images [1, 5, 7, 15, 33, 52, 60]. In addition, we present a *variation* of Concerto, augmented with an additional set of 50k point clouds with 200k corresponding images lifted from scene videos [61] via feed-forward reconstruction [43], tailored for video-based spatial understanding. We also introduce an *interlude* of Concerto: a learned translator that linearly projects self-supervised representations into CLIP's language space [27, 32], enabling open-world perception.

The intersection of 2D and 3D self-supervised learning in Concerto yields a powerful synergy, enabling the emergence of superior spatial representations. PCA-colored visualizations reveal that Concerto captures more coherent and informative spatial features than SOTA 2D or 3D self-supervised models trained on a single modality (see Fig. 1). Concerto exceeds its predecessor, Sonata, with a 4.8% improvement in linear probing, achieving 77.3% mIoU on ScanNet semantic segmentation using a single linear layer. Notably, this performance also surpasses the concatenation of Sonata and DINOv2 (1.4%), demonstrating that the multisensory synergy in Concerto exceeds the representational upper bound achievable by single-modality self-supervised learning. With full fine-tuning, Concerto achieves SOTA performance across a range of scene perception tasks. For example, reaching 80.7% mIoU on ScanNet semantic segmentation.

| Semantic Segmentation | | | ScanNet Val [15] | | | ScanNet200 Val [35] | | |
|---|---|---|---|---|---|---|---|---|
| Method | Type | Encoder | mIoU | mAcc | allAcc | mIoU | mAcc | allAcc |
| DINOv2 [28] | 2D Image SSL | ViT-G | 63.09 | 75.50 | 82.42 | 27.42 | 37.59 | 72.80 |
| Sonata [49] | 3D Point SSL | PTv3-B | 72.52 | 83.11 | 89.74 | 29.25 | 41.61 | 81.15 |
| Sonata×DINOv2 | 3D SSL×2D SSL | Both | 75.91 | 85.36 | 91.25 | 36.67 | 46.98 | 82.85 |
| **Concerto** (ours) | 2D-3D Joint SSL | PTv3-B | **77.32** | **86.58** | **91.74** | **37.41** | **49.49** | **83.29** |

Table 1: **Linear probing results on 3D semantic segmentation.** We compare self-supervised features learned from 2D, 3D, their feature concatenation, and our 2D-3D joint SSL model, Concerto (as a preview). Notably, the concatenation of 2D and 3D features outperforms either modality alone, suggesting that the two modalities encode complementary information. Concerto achieves the best performance across all metrics, demonstrating its ability to learn superior spatial representations.

## 2 Beyond Single Modality: Toward a New World of Representations

This section presents a pilot study to explore high-level questions surrounding self-supervised representations. These questions form the conceptual foundation of our research, and our methodology emerges as a natural and simple response to the insights gained here.

### 2.1 Is There a Superior Representation Space Beyond Single-Modality Learning?

Self-supervised learning on 2D images and 3D point clouds has achieved remarkable progress in visual representation learning. However, when trained independently, these models may capture only modality-specific perspectives of the spatial world. Just as a person who has only seen an *apple* but never tasted one may lack a sense of its flavor or texture, single-modal learning inevitably misses critical dimensions of the world. This raises a fundamental question: Is there a superior representation space that can emerge from the synergy between 2D and 3D modalities?

To probe this question, we begin with a simple pilot experiment: fusing self-supervised features from image and point cloud models, prior to any explicit learning of cross-modal synergy. Specifically, we select two representative self-supervised models trained independently on images (DINOv2 [28]) and point clouds (Sonata [49]). We lift image features into 3D space using depth and camera parameters, and concatenate them with point cloud representations to enable feature-level fusion. We benchmark the 2D, 3D, and fused representations via linear probing on 3D scene-level semantic segmentation using the ScanNet [15] dataset, with results presented in Tab. 1. Notably, this naive combination outperforms both individual modalities, suggesting the presence of complementary information and hinting at a richer representational space.

However, simply concatenating 2D and 3D self-supervised features, while yielding a stronger representation space, still falls short of uncovering the *unexplored new world* we are seeking. This approach lacks integration during learning and cannot fully capture the synergy that emerges when modalities are learned together. The deeper insight lies in the potential of multi-modal joint representation learning—not only to align complementary signals across modalities, but also to form coherent, predictive embeddings that generalize beyond their source. Ideally, such fused representations can be retrieved from a single modality, even if they were originally learned through multi-modal interaction. This form of joint 2D-3D representation learning is intuitive, as it mirrors how humans form concepts, as discussed in Sec. 1 and illustrated in Fig. 2.

*This insight leads to our methodology: designing a unified framework that learns to embed spatial information through both intra-modal refinement and cross-modal prediction.*

### 2.2 Can Multi-modal Self-Supervised Representations Speak the Language of Concepts?

Human language is often considered a compressed and symbolic interface to abstract concepts learned through multisensory synergy [6]. If multi-modal self-supervised representation learning succeeds in forming unified abstract concepts, then such representations should, in principle, be able to align with human language—perhaps even through a simple linear projection. This perspective raises a natural question: Can multi-modal self-supervised representations, learned entirely without human language, speak the language of concepts?

We believe the answer to this question is ultimately yes. However, our current study is grounded in the spatial domain, leveraging only 2D images and 3D point clouds. This limited sensory scope

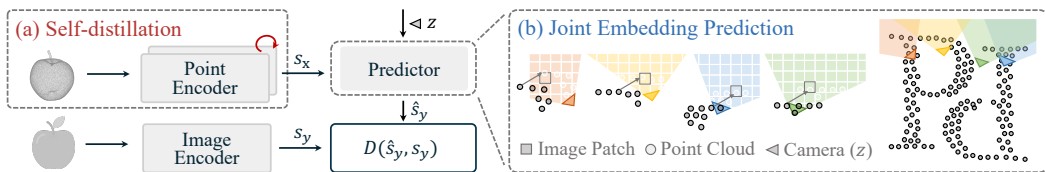

Figure 3: **Overview of the Concerto architecture.** Concerto simulates human multisensory synergy by coupling (a) intra-modal self-distillation on 3D point clouds to progressively refine its internal spatial representations (see Sec. 3.1), and (b) cross-modal joint embedding prediction that aligns point features with corresponding image patch features using camera parameters (see Sec. 3.2). The self-distillation branch (a) employs a restricted online clustering objective, while the joint embedding prediction (b) applies a looser cosine similarity constraint. This dual self-supervised objective encourages the emergence of coherent, modality-agnostic spatial representations.

makes it challenging to fully align with human language, which emerges from a far richer blend of modalities. Still, we propose that linearly probing self-supervised representations into a language embedding space, such as CLIP's, offers a meaningful way to evaluate progress toward this goal. Beyond serving as a diagnostic tool, this projection also extends the open-vocabulary capabilities of self-supervised spatial representations, offering a step toward broader concept grounding.

*We propose linear probing into the CLIP feature space as a next-level evaluation criterion for self-supervised learning beyond single modality.*

## 3 Concerto: Joint 2D-3D Self-Supervised Learning

This section introduces the joint 2D-3D self-supervised learning framework of Concerto. The proposed architecture is intentionally simple, designed to highlight the power of multisensory synergy through strong empirical performance. An overview of concerto architecture is present in Fig. 3.

### 3.1 Intra-Modal Self-Distillation

The primary duty of Concerto is to validate the superiority of multi-modal synergy, rather than seeking innovation in single-modality self-supervised learning architecture. For this reason, the intra-modal branch of Concerto builds upon the recent Sonata framework [49], which applies self-distillation [10, 28] to learn point cloud representations without supervision. We briefly revisit this architecture here and refer readers to the original Sonata paper for a detailed discussion on building a reliable self-supervised framework for sparse and unstructured point cloud data.

The intra-modal self-distillation in Concerto focuses solely on the 3D domain, where a Point Transformer V3 [47] is trained to produce stable and predictive features through a teacher-student paradigm. The student encoder is optimized to match the output of a momentum-updated teacher [18], using a clustering-based objective [9] that promotes consistency across augmented views of the same point cloud. A unique challenge in sparse point clouds is the geometric shortcut, where models collapse to easily accessible low-level geometric cues. These cues are not learned but introduced implicitly through the local kernel definitions of point cloud operators. Sonata mitigates this through several micro-designs that obscure explicit spatial signals and encourage learning from input features. This self-motivated refinement process allows the model to internalize geometric and structural priors from 3D data, forming the foundation for multi-modal learning in Concerto.

### 3.2 Cross-Modal Joint Embedding Prediction

We introduce an additional cross-modal self-supervised objective that continuously stimulates synergy from the images self-supervised representation into the point cloud domain. This design aligns with the core vision of LeCuns Joint Embedding Predictive Architecture (JEPA) [26], which advocates learning by predicting latent representations across modalities using a conditional predictor. The goal is to predict point cloud embeddings that match the associated pixel embeddings extracted from a self-supervised image encoder (e.g., DINOv2 [28]). Empirically, we find that cosine similarity provides the most effective criterion for training this predictive branch, and by applying strong point cloud data augmentations and exploring less aggressive image augmentations compared to DINOv2, Concerto learns more generalizable representations.

In our implementation, we divided scenes with a large number of images into several data pieces. Each one consists of one point cloud with 4 images. For scenes with fewer images (e.g., 5), we retain the original dataset divisions. As shown in Fig. 3, we respectively obtain point representations $s_x$ and image representations $s_y$ from the point encoder and image encoder. In the context of 2D images and 3D point clouds, our predictor takes camera parameters as the condition $z$ to establish correspondences between image pixels and point cloud points. The details of this process are provided in the Appendix. Then, for each image patch, we compute the mean of the features of points falling within it to get predicted patch features $\hat{s}_y$ from point clouds. Finally, we compute the loss $D(s_y, \hat{s}_y)$ using cosine similarity. This process introduces 2D self-supervised features into 3D self-supervised learning and stimulates the point intra-modal self-distillation process.

### 3.3 Synergy Emerged from Joint Self-Supervised Learning

The combination of intra-modal self-distillation and cross-modal joint embedding in Concerto emerges as a strong synergy, surpassing single-modality learning and elevating the interaction between self-supervised point cloud and image representations beyond naive fusion. The complementary cognitive signals from image self-supervised learning, through cross-modal joint embedding prediction, encourage point intra-modal self-distillation to extend capacity beyond a single modality and emerge a superior spatial representation surpassing the naive combination of self-supervised features from images and point clouds. This process yields spatial representations that are more expressive than those obtained by merely concatenating features from separate 2D and 3D models. We believe this chain reaction is central to the design of Concerto. Notably, the architecture also supports training on point clouds without paired images, enabling hybrid self-supervised learning without compromising scalability on large-scale 3D datasets.

## 4 Experiments

We evaluate Concerto's representations with various scene perception tasks using the same protocols as Sonata [49]. Specifically, linear probing, which keeps the encoder frozen and adapts the upcasted original scale features to downstream tasks; decoder probing, adding a lightweight decoder after the frozen encoder to facilitate adaptation; and full fine-tuning, where the encoder and decoder are optimized for downstream tasks. We further analyze Concerto's key properties with these results.

### 4.1 Main Results

**Semantic segmentation.** In Tab. 2a, we compared Concerto with the previous 3D encoder Sonata [49] on semantic segmentation tasks across multiple datasets: ScanNet [15], Scan-Net200 [35], ScanNet++ [52], and S3DIS Area5 [1] with full fine-tuning. Across all datasets, Concerto achieves SOTA performance with notable mIoU results, including 80.7% on ScanNet, 39.2% on ScanNet200, and 50.7% on ScanNet++. The most significant improvement is seen in the Scan-Net200 dataset, which contains 200 class categories. This suggests that while detecting fine-grained objects in sparse point clouds remains challenging, joint 2D-3D cross-modal learning enables Concerto to capture detailed semantic and geometry information, thus improving the models ability for such objects.

**Instance segmentation.** In Tab. 4, we further validate the robustness of Concerto across 4 widely recognized instance segmentation benchmarks. Concerto demonstrates the strongest performance in all evaluation methods. Notably, decoder probing on ScanNet outperforms full fine-tuning, suggesting that Concerto learns rich, generalizable representations during pretraining without task-specific adjustments. This demonstrates the advantage of leveraging general pretrained representations, which reduces the risk of distorting pretrained representations and overfitting in fine-tuning.

**Parameter efficiency.** In Tab. 2b, we demonstrate Concerto's parameter efficiency using the simplest linear probing and decoder probing across 4 semantic segmentation benchmarks. In particular, Concerto outperforms supervised learning using the PTv3 backbone [47] on all benchmarks with decoder probing. Even with linear probing, Concerto surpasses the supervised PTv3 on Scan-Net200 [35] and S3DIS [1]. Compared to Sonata [49], Concerto shows significant improvements on ScanNet200 [35] and ScanNet++ [52] with linear probing (+8.1% and +6.7% respectively). These results highlight a substantial improvement in scenes with larger numbers of classes.

**Data efficiency.** In Tab. 2c, we examine the data efficiency performance of Concerto on ScanNet Efficient Datasets [19] with limited scenes and annotations. Concerto outperforms Sonata [49] across all evaluation protocols. Notably, linear probing results surpass decoder probing and even full fine-

| Fine-Tuning Methods | Params Learn. | Pct. | ScanNet Val [15] mIoU | mAcc | allAcc | ScanNet200 Val [35] mIoU | mAcc | allAcc | ScanNet++ Val [52] mIoU | mAcc | allAcc | S3DIS Area 5 [1] mIoU | mAcc | allAcc |
|---|---|---|---|---|---|---|---|---|---|---|---|---|---|---|
| ○ SparseUNet [14] | 39.2M | 100% | 72.3 | 80.2 | 90.0 | 25.0 | 32.9 | 80.4 | 28.8 | 38.4 | 80.1 | 66.3 | 72.5 | 89.8 |
| ● PC [50] | 39.2M | 100% | 72.3 | 80.9 | 90.1 | 26.2 | 33.0 | 79.9 | 29.2 | 39.7 | 82.7 | 68.1 | 73.5 | 90.0 |
| ● CSC [19] | 39.2M | 100% | 72.8 | 81.0 | 90.7 | 26.9 | 33.7 | 80.6 | 32.5 | 41.1 | 83.7 | 70.7 | 76.4 | 90.8 |
| ● MSC [46] | 39.2M | 100% | 75.7 | 83.4 | 91.3 | 32.0 | 41.6 | 82.3 | 39.4 | 49.6 | 84.9 | 70.7 | 76.1 | 91.0 |
| ○ PTv3 [47] | 124.8M | 100% | 77.6 | 85.0 | 92.0 | 35.3 | 46.0 | 83.4 | 48.2 | 61.6 | 87.0 | 73.4 | 78.9 | 91.7 |
| ● MSC [46] | 124.8M | 100% | 78.2 | 85.3 | 92.2 | 33.4 | 43.7 | 83.4 | 48.7 | 61.9 | 87.2 | 69.9 | 74.9 | 91.2 |
| ● Sonata [49] | 124.8M | 100% | 79.4 | 86.1 | 92.5 | 36.8 | 46.5 | 84.4 | 49.3 | 62.4 | 87.6 | 76.0 | 81.6 | 93.0 |
| ● Concerto | 124.8M | 100% | **80.7** | **87.4** | **93.1** | **39.2** | **50.2** | **85.0** | **50.7** | **63.3** | **87.9** | **77.4** | **85.0** | **93.2** |

(a) Full fine-tuning. We evaluate Concerto using full fine-tuning, unlocking both encoder and decoder, and compare semantic segmentation mIoU, mAcc, allAcc(%) results across 4 benchmarks.

| Param. Efficiency Methods | Params Learn. | Pct. | ScanNet Val [15] mIoU | mAcc | allAcc | ScanNet200 Val [35] mIoU | mAcc | allAcc | ScanNet++ Val [52] mIoU | mAcc | allAcc | S3DIS Area 5 [1] mIoU | mAcc | allAcc |
|---|---|---|---|---|---|---|---|---|---|---|---|---|---|---|
| ○ SparseUNet [14] | 39.2M | 100% | 72.3 | 80.2 | 90.0 | 25.0 | 32.9 | 80.4 | 28.8 | 38.4 | 80.1 | 66.3 | 72.5 | 89.8 |
| ● PC [50] (lin.) | <0.2M | <0.1% | 5.6 | 9.7 | 50.0 | 0.5 | 0.9 | 40.3 | 1.8 | 3.1 | 46.4 | 11.4 | 18.6 | 52.3 |
| ● CSC [19] (lin.) | <0.2M | <0.1% | 12.6 | 18.1 | 64.2 | 1.3 | 2.1 | 53.0 | 2.8 | 4.5 | 53.6 | 24.4 | 32.0 | 66.4 |
| ● MSC [46] (lin.) | <0.2M | <0.1% | 14.1 | 20.3 | 62.9 | 1.5 | 2.5 | 53.6 | 4.5 | 6.6 | 61.3 | 27.9 | 35.5 | 71.1 |
| ○ PTv3 [47] | 124.8M | 100% | 77.6 | 85.0 | 92.0 | 35.3 | 46.0 | 83.4 | 48.2 | 61.6 | 87.0 | 73.4 | 78.9 | 91.7 |
| ● MSC [46] (lin.) | <0.2M | <0.2% | 21.8 | 32.2 | 65.5 | 3.3 | 5.5 | 57.5 | 8.1 | 11.9 | 64.7 | 32.1 | 42.4 | 70.9 |
| ● Sonata [49] (lin.) | <0.2M | <0.2% | 72.5 | 83.1 | 89.7 | 29.3 | 41.6 | 81.2 | 38.9 | 52.8 | 84.3 | 72.3 | 81.2 | 90.9 |
| ● Concerto (lin.) | <0.2M | <0.2% | **77.3** | **86.6** | **91.7** | **37.4** | **49.5** | **83.3** | **45.6** | **60.5** | **86.5** | **73.5** | **81.3** | **90.9** |
| ● Sonata [49] (dec.) | 16.3M | 13% | 79.1 | 86.6 | **92.7** | 33.5 | 44.5 | 84.1 | 45.2 | 57.4 | 86.8 | 74.5 | 80.4 | **92.6** |
| ● Concerto (dec.) | 16.3M | 13% | **79.5** | **87.6** | 92.6 | **37.8** | **50.5** | 84.1 | **48.3** | **62.3** | **87.7** | **75.5** | **84.2** | 92.3 |

(b) Parameter efficiency. By using linear probing (lin.) and decoder probing (dec.), we compare semantic segmentation mIoU, mAcc, allAcc(%) results across 4 benchmarks.

| Data Efficiency Methods | Limited Scenes (Pct.) 1% | 5% | 10% | 20% | Full | Limited Annotation (Pts.) 20 | 50 | 100 | 200 | Full |
|---|---|---|---|---|---|---|---|---|---|---|
| ○ PTv2 [45] | 24.8 | 48.1 | 59.8 | 66.3 | 75.4 | 58.4 | 66.1 | 70.3 | 71.2 | 75.4 |
| ○ SparseUNet [14] | 26.0 | 47.8 | 56.7 | 62.9 | 72.2 | 41.9 | 53.9 | 62.2 | 65.5 | 72.2 |
| ● CSC [19] | 28.9 | 49.8 | 59.4 | 64.6 | 73.8 | 55.5 | 60.5 | 65.9 | 68.2 | 73.8 |
| ● MSC [46] | 29.2 | 50.7 | 61.0 | 64.9 | 75.4 | 61.0 | 65.6 | 68.9 | 69.6 | 75.4 |
| ○ PTv3 [47] | 25.8 | 48.9 | 61.0 | 67.0 | 77.2 | 60.1 | 67.9 | 71.4 | 72.7 | 77.2 |
| ● PPT [48] (sup.) | 31.1 | 52.6 | 63.3 | 68.2 | 78.2 | 62.4 | 69.1 | 74.3 | 75.5 | 78.2 |
| ● Sonata [49] (lin.) | 43.6 | 62.5 | 68.6 | 69.8 | 72.5 | 69.0 | 70.5 | 71.1 | 71.5 | 72.5 |
| ● Sonata [49] (dec.) | 44.5 | 64.1 | 69.8 | 72.5 | 79.1 | 69.8 | 73.1 | 75.0 | 76.3 | 79.1 |
| ● Sonata [49] (f.t.) | 45.3 | 65.7 | 72.4 | 72.8 | 79.4 | 70.5 | 73.6 | 76.0 | 77.0 | 79.4 |
| ● Concerto (lin.) | **48.2** | **69.1** | 73.6 | 75.0 | 77.3 | **73.9** | 75.2 | 76.2 | 76.3 | 77.3 |
| ● Concerto (dec.) | 44.6 | 67.9 | 73.7 | 74.6 | 79.5 | 72.6 | 74.6 | 76.7 | 77.6 | 79.5 |
| ● Concerto (f.t.) | 46.5 | 69.0 | **75.3** | **76.1** | 80.7 | 73.3 | **76.7** | **77.6** | **78.4** | 80.7 |

(c) Data efficiency. We adopt the ScanNet Data Efficient [19] benchmark and compare the validation mIoU(%) results of Concerto with previous methods in three evaluation protocols.

Table 2: **Semantic segmentation.** We train Concerto on ScanNet [15], ScanNet++ [52], Structured3D [60], S3DIS [1], ArkitScenes [7], and HM3D [33] datasets, utilizing ScanNet, ScanNet200, ScanNet++, and S3DIS to evaluate the model by linear probing, decoder probing, and full fine-tuning and ScanNet Data Efficient [19] to evaluate the data efficiency. The pre-training setting is the default, described in Tab. 6. More specific pre-training details are available in the Appendix.

tuning (SFT) in extreme data-limited scenarios (1%, 5% limited scenes, and 20-point annotation per scene). This observation aligns with findings in the image domain [57], where linear probing outperforms full fine-tuning in out-of-distribution situations. In our case, when training on limited data, the whole evaluation dataset becomes an out-of-distribution situation. This significant emerging property reveals two key insights: more generalizable representations and more efficient adaptation potential. This could signal a potential shift toward Low-Rank Adaptation (LoRA) methods [20] for fine-tuning point cloud backbones. Detailed LoRA fine-tuning results are provided in the Appendix.

| Scale | ScanNet Val | | | ScanNet200 Val | | | ScanNet++ Val | | | S3DIS Area 5 | | |
|---|---|---|---|---|---|---|---|---|---|---|---|---|
| Model Size | mIoU | mAcc | allAcc | mIoU | mAcc | allAcc | mIoU | mAcc | allAcc | mIoU | mAcc | allAcc |
| 16M(T) | 67.7 | 78.5 | 87.4 | 24.9 | 34.4 | 79.3 | 33.7 | 45.9 | 82.7 | 65.2 | 73.6 | 88.6 |
| 39M(S) | 76.6 | 86.6 | 91.5 | 34.4 | 46.3 | 83.1 | 43.1 | 57.6 | 86.2 | 71.3 | 80.4 | 90.1 |
| 108M(B) | 77.3 | 86.6 | 91.7 | 37.4 | 49.5 | 83.3 | 45.6 | 60.5 | 86.5 | 73.5 | 81.3 | 90.9 |
| 207M(L) | 77.5 | 86.6 | 92.1 | 38.6 | 49.8 | 83.9 | 46.3 | 59.9 | 86.7 | 73.7 | 81.4 | 91.1 |

Table 3: **Scaling Up.** Model T, S, B is trained on point cloud datasets, while Model L is trained on point cloud datasets and an additional video dataset.

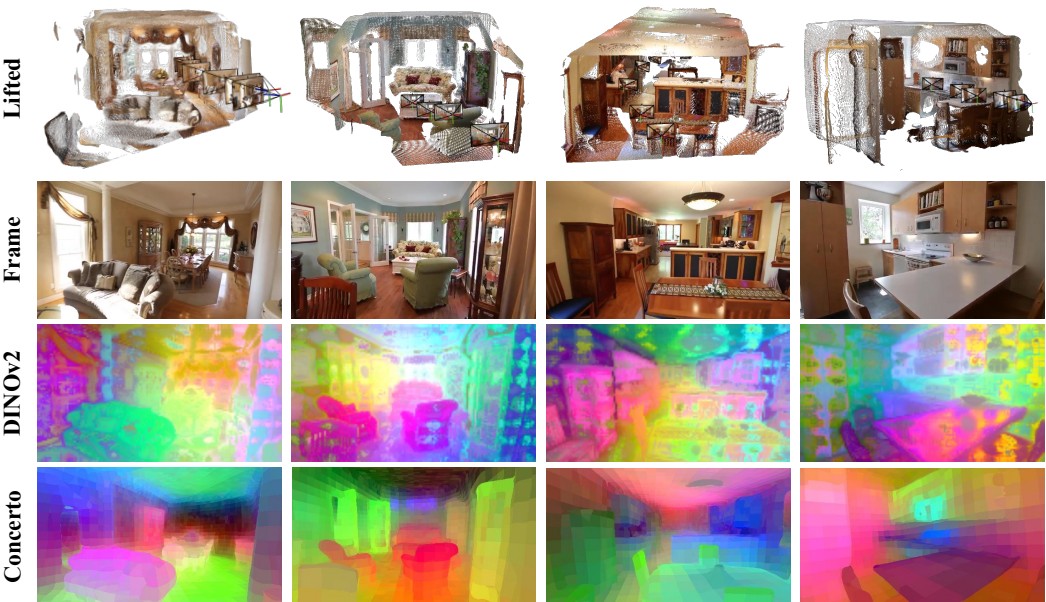

Figure 4: **Video spatial perception.** Concerto can be directly applied to video-lifted data (top row). The PCA visualizations (bottom two rows) illustrate that Concerto learns more fine-grained and semantically consistent features compared to DINOv2.

**Video processing.** As shown in the Fig. 4, the variation of Concerto demonstrates strong adaptability to video-lifted data. We hypothesize that certain spatial-specific information is more effectively captured in the lifted space. By leveraging the current feed-forward reconstruction method VGGT [43] to reconstruct the point clouds from videos, we generate a diverse range of point cloud data. Incorporating these lifted point clouds into the pipeline allows Concerto to learn more generalizable representations, enhancing its ability for real-time video spatial perception. Moreover, by including video data into training datasets, we aim to extend the scaling ability of Concerto further. In Tab. 3, we provide linear probing results of different model sizes, where the large model variant trained with the additional video data demonstrates significant potential for further scaling. Details of this Concerto variation and the method of lifting are presented in the Appendix.

**Language probing.** Language develops from a fundamental understanding of the physical world, which motivates many works on language alignment with 3D knowledge as [21, 24]. In Tab. 5, we demonstrate Concerto's ability to formulate concepts similar to human language, paving the way for future exploration of alignment with text-based semantic spaces. With linear probing trained on the same datasets as the pretraining stage, we translate

| Language | ScanNet Val | | |
|---|---|---|---|
| Methods | mIoU | mAcc | allAcc |
| MSC [46] | 12.42 | 18.55 | 59.89 |
| Sonata [49] | 41.71 | 61.29 | 78.86 |
| Concerto | 44.56 | 64.76 | 80.76 |

Table 5: **Language probing.**

Concerto's representations to language space by aligning LSeg [27] image encoder output to our linear probing output. Without ground truth labels, Concerto achieves 44.56% mIoU on ScanNet zero-shot segmentation. Although this lags behind the 77.3% mIoU by supervised linear probing,

| Ins. Seg. | Params | | ScanNet Val [15] | | | ScanNet200 Val [35] | | | ScanNet++ Val [52] | | | S3DIS Area 5 [1] | | |
|---|---|---|---|---|---|---|---|---|---|---|---|---|---|---|
| Methods | Learn. | Pct. | mAP$_{25}$ | mAP$_{50}$ | mAP | mAP$_{25}$ | mAP$_{50}$ | mAP | mAP$_{25}$ | mAP$_{50}$ | mAP | mAP$_{25}$ | mAP$_{50}$ | mAP |
| ○ PTv3 [47] | 124.8M | 100% | 77.5 | 61.7 | 40.9 | 40.1 | 33.2 | 23.1 | 46.3 | 39.6 | 25.4 | 55.7 | 49.4 | 37.8 |
| ● MSC [46] (lin.) | <0.2M | <0.2% | 13.3 | 5.3 | 2.3 | 2.3 | 1.0 | 0.4 | 4.8 | 2.6 | 1.3 | 19.0 | 13.0 | 9.7 |
| ● Sonata [49] (lin.) | <0.2M | <0.2% | 72.6 | 53.9 | 30.7 | 30.9 | 21.3 | 10.9 | 34.6 | 26.5 | 14.8 | 45.8 | 36.6 | **26.1** |
| ● Concerto (lin.) | <0.2M | <0.2% | **75.4** | **55.7** | **31.1** | **38.2** | **27.7** | **14.9** | **40.1** | **31.3** | **18.2** | **52.5** | **40.8** | 25.7 |
| ● Sonata [49] (dec.) | 16.3M | 13% | 76.8 | 62.8 | 40.8 | 40.8 | 33.3 | 22.8 | 39.4 | 33.5 | 22.7 | 63.7 | 57.1 | 45.1 |
| ● Concerto (dec.) | 16.3M | 13% | **81.1** | **64.2** | **42.7** | **41.8** | **34.4** | **24.0** | **42.2** | **35.3** | **23.4** | **66.8** | **58.1** | 45.1 |
| ● MSC [46] (f.t.) | 124.8M | 100% | 78.4 | 62.9 | 41.1 | 40.5 | 33.8 | 23.4 | - | - | - | 56.3 | 50.5 | 38.1 |
| ● PPT [48] (sup.) | 124.8M | 100% | 78.9 | 63.5 | 42.1 | 40.8 | 34.1 | 24.0 | - | - | - | 57.5 | 51.2 | 39.7 |
| ● Sonata [49] (f.t.) | 124.8M | 100% | 79.2 | 63.9 | 42.4 | 42.1 | 35.6 | 25.4 | 43.3 | 36.5 | 24.6 | 63.8 | 57.4 | 45.5 |
| ● Concerto (f.t.) | 124.8M | 100% | **79.5** | **64.9** | **42.9** | **45.8** | **38.7** | **27.4** | **44.3** | **38.3** | **26.0** | **67.5** | **61.0** | **46.4** |

Table 4: **Instance segmentation.** Concerto demonstrates the strongest performance for instance segmentation across 4 datasets with all evaluation protocols.

| cross intra | o. c. o. c. | c. s. o. c. | o. c. c. s. | c. s. c. s. |
|---|---|---|---|---|
| lin. | 60.7 | **75.6** | 31.6 | 74.7 |
| dec. | 72.0 | **78.6** | 66.0 | 78.3 |

(a) **Criteria type.** o.c.: online clustering criteria in DINOv2 [28]; c.s.: cosine similarity.

| image usage | 0% | 20% | 50% | 70% | 100% |
|---|---|---|---|---|---|
| lin. | 70.9 | 73.5 | 75.2 | 75.4 | **75.6** |
| dec. | 77.4 | 77.4 | 77.8 | 78.4 | **78.6** |

(b) **Image usage.** We control the image usage ratio to present the effectiveness of joint cross-modal learning.

| criteria weight | 1:2 | 2:2 | 3:2 |
|---|---|---|---|
| lin. | 75.6 | **76.1** | 75.7 |
| dec. | 78.6 | 78.6 | **78.8** |

(c) **Criteria weight.** A suitable weight ratio (cross:intra) is needed to balance intra- and cross-modal components.

| img. aug. | w | w/o |
|---|---|---|
| lin. | **76.7** | 75.6 |
| dec. | **78.6** | 78.6 |

(d) **Image augmentation.** Weak image augmentation shows a positive impact.

| vis. points | ×1 | ×2 |
|---|---|---|
| lin. | **75.6** | 75.5 |
| dec. | **78.6** | 78.1 |

(e) **Visible points.** Fewer visible points, better performance. ×1: 65536 points.

| upcast level | 2 | 3 | 4 |
|---|---|---|---|
| lin. | 75.1 | **75.6** | 75.3 |
| dec. | 78.5 | **78.6** | 78.2 |

(f) **Upcast level.** Upcast level, processed as [49], refers to the concatenation level of features in cross-modal learning here.

| data scale | 23k | 40k |
|---|---|---|
| lin. | 75.6 | **76.6** |
| dec. | 78.6 | **79.2** |

(g) **Data scale.** 23k is the default ablation setting and the total dataset contains 40k.

| model scale | s | b |
|---|---|---|
| lin. | 76.6 | **77.3** |
| dec. | 79.2 | **79.5** |

(h) **Model scale.** Training on 40k. s: 39M backbone; b: 108M backbone.

Table 6: **Ablation study.** The default ablation setup trains on ScanNet [15] and Structured3d [60] with 39M PTv3 [47] model as in Sonata [49]. If not specified, other default settings are in the Appendix. For Tab. 6g and Tab. 6h, we scale the setup to match the model used in the main results. All of our designs are enabled by default. Default settings are marked in  blue .

we expect that further language-conditioned probing will yield comparable results, marking a significant step toward bridging 3D spatial representations with text.

## 4.2 Ablation Study

In this section, we ablate intriguing properties for Concerto in Tab. 6 with the default setting in captions, evaluating the ablation on ScanNet semantic segmentation using linear and decoder probing.

**Joint cross-modal learning.** In Tab. 1, we investigate the influence of joint cross-modal learning by comparing Concerto with strong baseline models DINOv2 and Sonata. Concerto outperforms both and surpasses their native feature concatenation. These results demonstrate that joint cross-modal learning does more than merely merge information from different modalities; it enables the model to learn richer emerging representations that were previously unattainable.

**Criteria type.** In Tab. 6a, we show that using cosine similarity as the loss function in the cross-modal joint embedding prediction component and cross-entropy-based online clustering loss from DINOv2 [28] in the self-distillation component facilitates joint 2D-3D self-supervised learning in latent space. This combination reduces strict constraints and minimizes conflicts between cross-modal and intra-modal learning, enabling a smoother joint learning process for the two objectives.

**Image usage.** In Tab. 6b, we further investigate the effect of multisensory interactions by varying the input ratio of point clouds with images. The results show that even with a small ratio, such as 20%,

joint cross-modal learning is effective, leading to improvements in linear probing. When the image usage ratio reaches 50%, the linear probing result is comparable to that with 100% image usage, while decoder probing continues to show potential for further improvement. These results suggest that shallow linear representations are easier to discover with a smaller proportion of images, while deeper representations, which require decoder probing, benefit from a higher image usage ratio.

**Criteria weight.** In Tab. 6c, we observe that the criteria weight ratio between cross-modal and intra-modal components affects performance. Given the distinct objectives of intra-modal and cross-modal learning, maintaining a balanced loss-weight ratio is essential for optimal performance. The 2:2 ratio outperforms the others in linear probing.

**Image augmentation.** Data augmentations are crucial in self-supervised learning. As Sonata [49] already explores point cloud augmentations in self-distillation, we focus on the cross-modal image augmentations here. Initially, we follow DINOv2 strong data augmentations, which results in a lower performance with a linear probing mIoU of 75.27%. However, when we apply less aggressive augmentations, the performance surpasses our default setting without image augmentations, as in Tab. 6d. The details of the image augmentations are provided in the Appendix. Since the image encoder here is frozen, it does not benefit from data augmentations. Additionally, overly aggressive image augmentation may confuse the point encoder with excessive distortions. Thus, careful selection of image augmentations is essential. Currently, our model does not apply image augmentations, and we plan to explore this in future updates.

**Visible points.** In Tab. 6e, we investigate the impact of visible points in the point cloud from image views. We hypothesize that while a large amount of matched point-pixel pairs can offer more complete information, the smaller number of pairs forces the model to predict across modalities for the surrounding context. As the task becomes more challenging, the model is encouraged to dig deeper into semantics, leading to better performance. In ablation, the performance is quite similar as the point numbers we selected are still quite small, compared with the number of image pixels and points in point clouds.

**Upcast level.** While Sonata [49] studies the influence of upcast levels on self-supervised learning representations, we present the performance of different cross-modal feature upcast levels in Tab. 6f. The model at upcast level 3 achieves the best performance, indicating that the upcast level 3 is close to the corresponding scales of the image and point cloud. Additionally, level 3 outperforms level 4 as level 4 may retain too many low-level details that are not beneficial for joint embedding learning. Furthermore, level 3 surpasses level 2, as the use of Sonata's self-distillation technique at level 2 might introduce conflicts between intra-modal self-supervised learning and cross-modal joint learning of the same upcast level, ultimately leading to negative effects.

**Data scale and model weight.** Additionally, following the approach of Sonata [49], we scale our datasets from 23k to 40k, resulting in a significant improvement shown in Tab. 6g. Likewise, aligning our model size with that of Sonata (108M) further enhances the performance, as demonstrated in Tab. 6h. Larger datasets provide more diverse and comprehensive information, enabling the model to learn more general patterns. As the model size increases, it becomes better equipped to capture complex relationships. Combining both naturally leads to more generalized representation outputs.

## 5 Related Work

**2D image self-supervised learning.** Aimed at utilizing oceans of unlabeled data, image self-supervised learning has seen significant progress [3, 12, 18, 28, 56]. These methods always focus on learning invariant representations through transformations or augmentations of the data. One of the most notable achievements in this field is DINOv2 [28], producing high-quality image representations. Building on the success of DINOv2, Concerto extends its reliable 2D image representations to the cross-modal domain, incorporating both 2D image and 3D point cloud data for superior representations in the 3D domain.

**3D point self-supervised learning.** While self-supervised learning has made significant progress in the image domain, it is still in the starting stage of 3D point clouds. Building on the success of Sonata [49], we further extend the previous works [42, 46, 50] on unimodal self-supervised learning with scene-level data to joint 2D-3D self-supervised learning for superior representation extracting ability. Before Sonata [49], most of the point self-supervised learning works suffer from geometry shortcuts due to the sparse and unordered nature of point clouds. Based on its predecessor, Concerto includes 2D images in its pipeline: leveraging point clouds to predict image features from DI-

NOv2 [28] by cross-modal joint embedding prediction and including 200k video-lifted point clouds by feed-forward reconstruction methods [43] into training datasets.

**Spatial understanding with joint 2D-3D data.** With the rapid advancements in 2D self-supervised learning and its remarkable performance, many methods for point cloud representation now incorporate image features into their pipelines. Approaches such as lifting projections [16, 39], differentiable rendering [25, 54, 58], direct distillation [11, 13, 55, 62], attention-based feature fusion [59], and using text-aligned image encoders for open-vocabulary tasks [22, 30, 40, 44] aim to incorporating image features in 3D learning. However, these methods primarily focus on imitating image features in point cloud representations, often overlooking the full potential of multi-modal interaction. Recently, Locate3D [2] seeks to develop generalizable representations beyond 2D image features but still relies on 2D image features during inference with a complicated pipeline. In contrast, Concerto utilizes joint 2D-3D embedding prediction during training, resulting in unified and rich representations beyond individual 2D or 3D features and their simple combination, enabling superior representations in inference with only point clouds.

## 6 Conclusion and Discussion

In this work, we present Concerto, achieving SOTA performance across multiple benchmarks. Additionally, we present the variation of Concerto for video spatial perception and the interlude of Concerto, which explores potential future alignment with text spaces. Concerto holds great promise for joint multi-modal self-supervised learning. Currently, it excels as a joint 2D-3D self-supervised learning model in the 3D domain, delivering superior performance in spatial representation learning. However, our goal extends beyond this. We discuss limitations and future works as follows:

- *Native multi-modal representation learning.* Our current training recipe freezes the image encoder, treating it as a static feature extractor. A compelling future direction is to unfreeze both the image and point cloud encoders for joint native multi-modal representation pre-training. This approach would enable the two modalities to mutually enhance one another during learning, fostering the development of a more robust shared representation space.
- *Deep semantic grounding of language in point clouds.* While our work Concerto employs linear probing as an effective metric for evaluating feature quality, this method deliberately promotes a shallow alignment between point clouds and language to avoid the influence of post-training part on the evaluation performance of pretraining features. For real-world applications, this is insufficient to just have a shallow language alignment with specific key terms. A critical next step is to develop architectures and training objectives that move beyond simple feature alignment towards deep semantic grounding. The goal is to enable the learned representations to comprehend and respond to nuanced, indirect, or compositional linguistic descriptions, which remains a significant open challenge.
- *Unified self-supervised learning paradigm for diverse point cloud domains.* Self-supervised learning for point clouds has historically been fragmented, with models tailored to specific domains (e.g., indoor, outdoor, object-level) to handle their distinct characteristics of scale and density. We believe that a unified pre-training paradigm trained on the data from different domains can produce more powerful and generalizable representations.By incorporating varied data sources like lidar point clouds, video-lifted point clouds, object-centric point clouds, and dynamic egocentric point clouds, a single self-supervised model can learn features that are robust to domain shifts. This enhanced generalization is expected to significantly boost performance on a wide array of downstream tasks, even those confined to a single domain.

**Acknowledgments.** The research presented in this paper was supported by the National Natural Science Foundation of China (No. 62422606, 62201484).

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

Figure 5: **Qualitative visualization.** Concerto performs well across different point cloud inputs: a complete scene (top two rows) and an incomplete scene (bottom two rows).

# Appendix

Concerto is a superior spatial representation point encoder capable of handling a wide range of scene types, including those with varying completeness in Fig. 5, video-lifted point clouds in Fig. 6, and the large scene in Fig. 7. Here, we further present the detailed implementation and results.

# A    Additional Implementation

We adopt the detailed parameters from Sonata [49] for intra-modal self-distillation and refer readers to the original Sonata paper for an in-depth description of its implementation. In this section, we provide a thorough explanation of the implementation for cross-modal joint embedding prediction.

## A.1    Combination of Intra-Modal and Cross-Modal Learning

As Sonata, we use 4 local views, 2 masked views, and 2 global views, with the first global view serving as the principal view. For cross-modal joint embedding prediction, we utilize the representations from the first masked view (based on the principal view) to predict the corresponding image representations. The cross-modal cosine similarity loss is computed at upcast level 3, while the online clustering cross-entropy loss for intra-modal self-distillation is calculated at upcast level 2.

## A.2    Correspondence Between Pixels and Points

To establish reliable 3D point to 2D pixel correspondences across camera views, we employ a two-step approach: 3D-to-2D projection followed by depth-based visibility verification.

Let $\boldsymbol{p} = (X, Y, Z)^T$ denote a 3D point in world coordinates. Each camera $c$ is defined by intrinsic matrix $\boldsymbol{K}$ and extrinsic matrix $[\boldsymbol{R}|\boldsymbol{t}]$. The standard pinhole camera model projects the 3D point $\boldsymbol{p}$ to 2D pixel coordinates $(x, y)$ and a projected depth $d_{\text{proj}}$:

$$d_{\text{proj}} \begin{bmatrix} x \\ y \\ 1 \end{bmatrix} = \boldsymbol{K}[\boldsymbol{R}|\boldsymbol{t}] \begin{bmatrix} X \\ Y \\ Z \\ 1 \end{bmatrix}. \tag{1}$$

To account for occlusions, we perform a visibility check comparing $d_{\text{proj}}$ with the depth value $d_c = \boldsymbol{D}_c(x, y)$ retrieved from camera $c$'s depth map $\boldsymbol{D}_c$ at the projected pixel coordinate $(x, y)$.

| Dataset | Source | Train | Val | Test | All |
|---|---|---|---|---|---|
| ScanNet [15] | real | 26,428 | 7,354 | 2,877 | 36,659 |
| ScanNet++ [52] | real | 49,315 | 1,583 | 1,208 | 52,106 |
| S3DIS [1] | real | 10,977 | 3,668 | 0 | 14,645 |
| ArkitScenes [7] | real | 72,481 | 9,786 | 0 | 82,267 |
| HM3D [33] | real | 64,936 | 8,240 | 0 | 73,176 |
| Structured3D [60] | synthesis | 65,160 | 6,722 | 6,396 | 78,278 |
| RE10K [61] | real | 166,680 | 0 | 18,464 | 185,144 |
| Concerto (ours) | mixed | **455,972** | **37,353** | **28,945** | **522,270** |

Table 7: **Image Data Source Collection.**

| Dataset | Source | Train | Val | Test | All |
|---|---|---|---|---|---|
| ScanNet [15] | real | 1,201 | 312 | 100 | 1,613 |
| ScanNet++ [52] | real | 856 | 50 | 50 | 956 |
| S3DIS [1] | real | 204 | 68 | 0 | 272 |
| ArkitScenes [7] | real | 4,498 | 549 | 0 | 5,047 |
| HM3D [33] | real | 8,117 | 1,030 | 0 | 9,147 |
| Structured3D [60] | synthesis | 16,635 | 1,722 | 1,648 | 20,005 |
| RE10K [61] | real | 41,670 | 0 | 4,612 | 46,282 |
| Concerto (ours) | mixed | **74,894** | **3,785** | **6,459** | **85,138** |

Table 8: **Point Cloud Data Source Collection.**

The point is considered visible if:

$$|d_c - d_{\mathrm{proj}}| < \epsilon_{\mathrm{depth}}, \tag{2}$$

where $\epsilon_{\mathrm{depth}}$ is set to 0.01 in our experiments. Additionally, the correspondence is rejected if $(x, y)$ falls outside image bounds or $D_c(x, y)$ contains invalid depth. This visibility check establishes a mapping between 3D points and corresponding 2D pixels, enabling direct correspondence between 3D points and ViT patches for cross-model joint embedding prediction mechanisms. Depending on the dataset, the depth map $D_c$ is obtained in different ways:

- **RGBD datasets.** Depth maps are directly available as the depth channel of RGBD images, such as Structured3D [60].
- **Known ground truth mesh.** For datasets like ScanNet [15], ScanNet++ [52], S3DIS [1], and ARKitScenes [7], depth maps are rendered from the ground truth 3D mesh using camera parameters.
- **Pixel-aligned point clouds.** For video-lifted point clouds (e.g., using VGGT [43] on RealEstate10K [61]), per-view depth maps $D_i$ are generated alongside point clouds $\mathcal{P}_i$. A point $p \in \mathcal{P}_i$ from camera $i$ can be visible from camera $j$ if it passes the visibility check.

For HM3D [33], which does not provide the raw images, we leverage Habitat-Sim [31] to simulate the scenes. For each navigatable room, we capture four images around the room with random initial camera orientations. The angular difference between consecutive images is 90 degrees. We record the camera parameters to compute the correspondence between points and pixels, as described previously. The total collections of our training data are shown in Tab. 7 and Tab. 8.

### A.3 Image Augmentations

We implement the same point cloud augmentations as Sonata. For image augmentations, we initially adopt the process from DINOv2 [28], excluding geometric augmentations to simplify the alignment between pixels and points. Specifically, we apply color jittering, random grayscale, and Gaussian blur to the images, consistent with the settings used in DINOv2. This results in a slight drop in the mIoU on ScanNet semantic segmentation to 75.27%, compared to using the original images. Consequently, we continue to explore more suitable image augmentations. In the ablation study, we apply random color jittering, with the same intensity as the point cloud augmentations, along with Gaussian blur. This weaker augmentation improves Concerto's performance, which is expected since the image encoder is currently frozen. Stronger augmentations may yield better results once both the image and point branches are unlocked for joint learning.

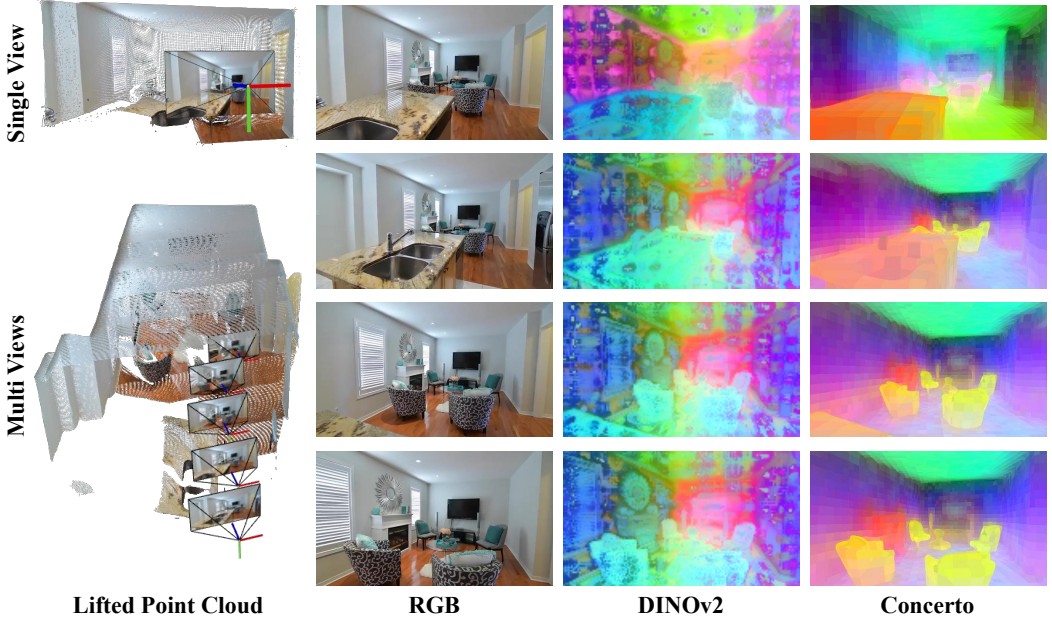

| Lifted Point Cloud | RGB | DINOv2 | Concerto |
| --- | --- | --- | --- |

Figure 6: **Video perception.** Concerto can be applied to single-view (top row) and multi-view video-lifted data (bottom three rows). We visualize the PCA of one video in RE10K [61]. In the multi-view setting, the representations from all the frames are computed together for consistency.

## A.4 Experimental Setting

**Software and hardware environment.**

- CUDA version: 12.4
- PyTorch version: 2.4.1
- Python version: 3.10.15
- GPU: Nvidia H20 $\times$ 16 for pretraining; Nvidia H20 $\times$ 8 for evaluation.
- CPU: $\times$ 360 for pretraining; $\times$ 180 for evaluation.
- Memory: 3600GB for pretraining; 1800GB for evaluation.
- Time: 85h for pretraining of base model without video data.

**Data license.** We use the open-source datasets ScanNet [15], ScanNet++ [52], S3DIS [1], Structured3D [60], ARKitScenes [7], Habitat Matterport3D [33] and RealEstate10K [61] in latest versions. S3DIS, Structured3D, ScanNet, and ScanNet++ have custom licenses. RealEstate10K is licensed by Google LLC under a Creative Commons Attribution 4.0 International License. ARKitScenes is licensed by Apple Inc. HM3D is licensed by Matterport.

**Training details.** For pretraining, we leverage all train, val, and test splits to train the self-supervised model. For evaluation with linear probing, decoder probing, and full fine-tuning, we train on the train split and test on the val split of ScanNet, ScanNet++, ScanNet200, and Area 5 of S3DIS. We use AdamW as the optimizer, and cosine annealing policy as the scheduler. The learning rate is adjusted with the encoder depth, and the max one is 0.004. The pretraining epoch is 100. For cross-modal joint embedding prediction, we set DINOv2 image encoder input resolution 518$\times$518.

## B Additional Results

### B.1 Concerto with Video-Lifted Point Clouds

We utilize the current feed-forward reconstruction model VGGT [43] to lift RealEstate10K [61] video data to point clouds. Based on the camera poses, we heuristically select video clips with larger camera pose transforms in comparison and abandon those with smaller camera pose transforms. With these video clips, we can build a video dataset with more completed scenes. In Fig. 6, we utilize Concerto to deal with single-view lifted data and multi-view lifted data. The visualizations

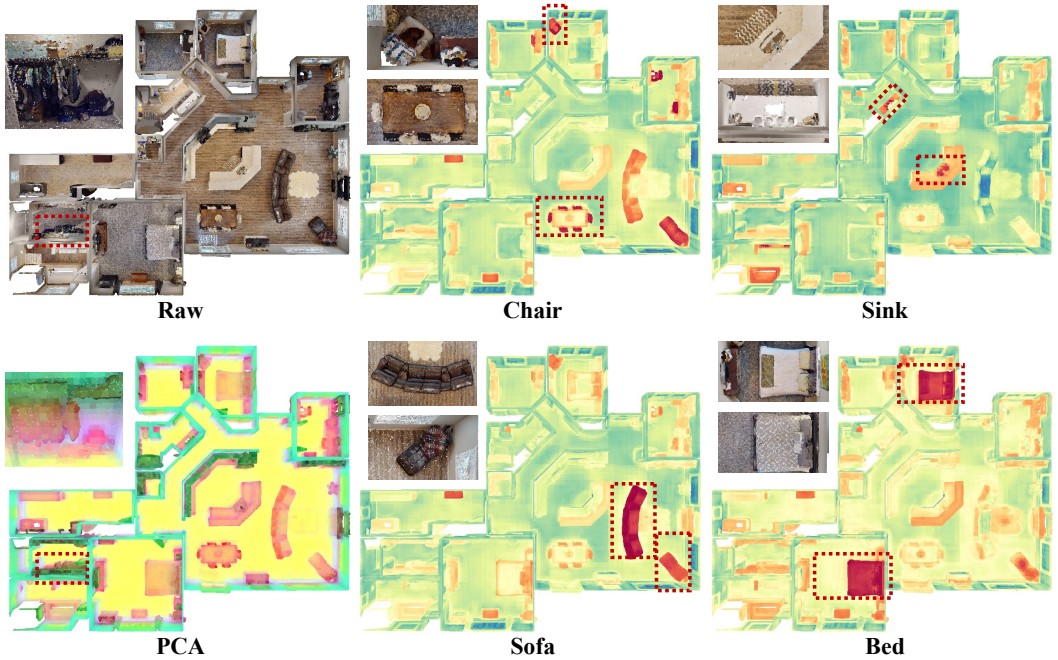

Figure 7: **Language locate.** We visualize the PCA of a large house scene from HM3D [33] along with the heatmap of zero-shot language-based object localization results. The upper-left part of the scene shows detailed local information. Given specific words, Concerto with text-aligned linear probing successfully locates objects in a zero-shot setting.

| Model | ScanNet Val | | | ScanNet200 Val | | | ScanNet++ Val | | | S3DIS Area 5 | | |
|---|---|---|---|---|---|---|---|---|---|---|---|---|
| img. enc. | mIoU | mAcc | allAcc | mIoU | mAcc | allAcc | mIoU | mAcc | allAcc | mIoU | mAcc | allAcc |
| DINOv2 (lin.) | **77.3** | **86.6** | **91.7** | **37.4** | **49.5** | **83.3** | 45.7 | 60.5 | 86.5 | **73.5** | **81.3** | 90.9 |
| SigLIP2 (lin.) | 76.3 | 86.0 | 91.4 | 36.7 | 48.9 | 82.7 | **45.8** | **61.4** | **86.8** | 72.3 | 79.4 | **91.0** |
| RADIO (lin.) | 73.5 | 84.0 | 90.3 | 31.0 | 42.3 | 81.8 | 42.7 | 57.2 | 85.3 | 72.9 | 80.5 | 90.9 |
| DINOv2 (dec.) | **79.5** | **87.6** | **92.6** | **37.8** | **50.5** | **84.1** | 48.3 | 62.3 | 87.7 | 75.5 | **84.2** | 92.3 |
| SigLIP2 (dec.) | 78.8 | 87.0 | 92.4 | 37.5 | 47.7 | 83.7 | 46.8 | 58.1 | 87.0 | 73.6 | 79.8 | 91.3 |
| RADIO (dec.) | 77.9 | 85.7 | 92.3 | 33.9 | 44.6 | 83.4 | 44.9 | 56.5 | 86.2 | 74.8 | 81.2 | 92.2 |
| DINOv2 (full.) | **80.7** | **87.4** | **93.1** | **39.2** | 50.2 | **85.0** | 50.7 | **63.3** | 87.9 | **77.4** | **85.0** | **93.2** |
| SigLIP2 (full.) | 79.7 | 86.9 | 92.7 | 38.4 | 49.9 | 83.9 | 50.0 | 62.0 | 88.2 | 75.0 | 80.2 | 92.5 |
| RADIO (full.) | 79.6 | 86.6 | 92.7 | 36.1 | 46.9 | 83.8 | 48.4 | 60.6 | 88.2 | 75.1 | 80.5 | 92.8 |

Table 9: **Segmentic segmentation of Concerto with different image encoders.** Concerto with DINOv2 based on self-distillation has the best performance in general.

show that Concerto adapts well to these two situations, suggesting that Concerto cannot only be applied to the offline video reconstruction but also to the single-view forward situation.

## B.2   Concerto with Language Probing

We leverage a simple linear layer to translate the representations from Concerto to CLIP's text space. During training, we force the linear probing output to align with the LSeg [27] image encoder's output, which does not need the ground truth labels to supervise. In the aligning process, we do not use masks and crop augmentations. The visualization results are shown in Fig. 7.

## B.3   Results with Different 2D Encoder

In this section, we compare the performance of different strong image encoders: DINOv2 [28], SigLIPv2 [41], and RADIO [34]. We adopt DINOv2 L version with a resolution of 518×518, SigLIPv2 So400m version with a patch size of 16 and resolution 512×512, and RADIOv2.5 L version with a resolution of 768×768. For each model, we pretrain a variant of Concerto on 40k data,

| Data Efficiency | Params | | Limited Scenes (Pct.) | | | | | Limited Annotation (Pts.) | | | | |
|---|---|---|---|---|---|---|---|---|---|---|---|---|
| Methods | Learn. | Pct. | 1% | 5% | 10% | 20% | 100% | 20 | 50 | 100 | 200 | Full |
| Concerto (lin.) | 0.02M | 0.02% | 48.2 | 69.1 | 73.6 | 75.0 | 77.3 | 73.9 | 75.2 | 76.2 | 76.3 | 77.3 |
| Concerto (dec.) | 16.3M | 13.1% | 44.6 | 67.9 | 73.7 | 74.6 | 79.5 | 72.6 | 74.6 | 76.7 | 77.6 | 79.5 |
| Concerto (full.) | 124.8M | 100.0% | 46.5 | 69.0 | **75.3** | 76.1 | 80.7 | 73.3 | 76.7 | 77.6 | 78.4 | 80.7 |
| Concerto (lora) | 0.3M | 0.2% | **48.4** | **70.2** | 74.9 | **76.8** | 79.8 | **75.1** | **77.2** | **78.3** | **78.7** | 79.8 |

Table 10: **Parameter Efficiency with LoRA.** Concerto with LoRA significantly improves the performance with a minimal number of learnable parameters, highlighting the reliability of pretrained Concerto representations and the effectiveness of LoRA fine-tuning.

| LoRA | Params | | ScanNet Val | | | ScanNet200 Val | | | ScanNet++ Val | | | S3DIS Area 5 | | |
|---|---|---|---|---|---|---|---|---|---|---|---|---|---|---|
| Methods | Learn. | Pct. | mIoU | mAcc | allAcc | mIoU | mAcc | allAcc | mIoU | mAcc | allAcc | mIoU | mAcc | allAcc |
| Concerto (lin.) | <0.2M | <0.2% | 77.3 | 86.6 | 91.7 | 37.4 | 49.5 | 83.3 | 45.6 | 60.5 | 86.5 | 73.5 | 81.3 | 90.9 |
| Concerto (dec.) | 16.3M | 13.1% | 79.5 | 87.6 | 92.6 | 37.8 | 50.5 | 84.1 | 48.3 | 62.3 | 87.7 | 75.5 | 84.2 | 92.3 |
| Concerto (full.) | 124.8M | 100.0% | **80.7** | 87.4 | **93.1** | **39.2** | 50.2 | **85.0** | **50.7** | **63.3** | **87.9** | **77.4** | **85.0** | **93.2** |
| Concerto (lora) | <0.5M | <0.5% | 79.8 | **87.9** | 92.7 | 38.4 | **51.9** | 84.1 | 47.3 | 60.8 | 87.7 | 75.5 | 81.4 | 92.6 |

Table 11: **Semantic segmentation with LoRA.** We compare the LoRA fine-tuning method on Concerto across four semantic segmentation benchmarks, demonstrating LoRA's remarkable capacity in general and the reliability of Concerto's original pretrained representations.

excluding video-lifted data. We evaluate these models across four datasets on semantic segmentation, as shown in Tab. 9. The results reveal that the Concerto model based on DINOv2, using self-distillation, achieves the highest mIoU in general. This suggests that in our joint self-supervised learning framework, the optimal synergy is achieved when representations from different domains are derived through intra-modal self-distillation. RADIO, which incorporates distilled information from multiple models, may damage the original self-distillation features from DINOv2, thus leading to a decrease in performance.

### B.4 Results with LoRA Finetuning

From the main results, we observe that linear probing outperforms full-finetuning in extreme data-scarce scenarios. This suggests that training methods may benefit from shifting toward LoRA-based fine-tuning. In this section, we present the results of LoRA fine-tuning. Specifically, we adapt LoRA to the point encoder and evaluate it with linear probing. We set the LoRA rank to 8, the LoRA alpha to 16, and the dropout rate to 0.1.

The results of ScanNet Data Efficiency are shown in Tab. 10. These results demonstrate that the LoRA-based method outperforms both linear probing and full fine-tuning in terms of mIoU across most scenarios, despite a small increase in learnable parameters compared to the original linear probing. This suggests that LoRA is an effective fine-tuning approach, particularly when data is limited. Notably, linear probing with LoRA achieves performance comparable to decoder probing in the full evaluation and only a 0.9% performance drop compared to full fine-tuning on mIoU, while offering significant improvements in training efficiency.

We also evaluate the LoRA fine-tuning on Concerto across four benchmarks, as shown in Tab. 11. The results demonstrate that LoRA fine-tuning shows performance comparable to decoder probing, even with relatively small learnable parameters. Overall, the LoRA fine-tuning demonstrates strong efficiency and performance across various benchmarks, highlighting two key insights: Concerto already yields reliable and generalizable representations, and leveraging pretrained representations combined with LoRA fine-tuning is both efficient and effective for further task adaptation.

## C  Broader Impact

In this work, we introduce Concerto, a powerful model for spatial representation learning, achieving SOTA performance in full fine-tuning and linear probing. Looking ahead, Concerto holds great promise for extending multi-modal learning beyond 2D-3D, benefiting downstream tasks such as autonomous driving, robotics, and mixed reality. However, if not properly trained, point cloud encoders may learn biases from the data, reinforcing societal stereotypes. Researchers should be aware of such potential negative social impacts and continuously monitor the training data for bias.

