# OpenReview forum: "Concerto: Joint 2D-3D Self-Supervised Learning Emerges Spatial Representations"
_NeurIPS.cc/2025/Conference — NeurIPS 2025 poster_

### Official Review · Reviewer_yNfS · 2025-06-04

**Clarity:** 4
**Significance:** 3
**Originality:** 3
**Rating:** 4
**Confidence:** 4

**Summary:**

This paper proposes the Concerto framework, which fuses image and point cloud modalities through 2D-3D joint self-supervised learning, and combines intra-modal self-distillation (3D point cloud feature refinement) with cross-modal joint embedding prediction (image-point cloud feature alignment) to achieve more consistent spatial representation learning. Experiments show that Concerto significantly outperforms single-modal methods on benchmarks such as ScanNet, with linear detection accuracy improved by 4.8%-14.2%, and achieves SOTA performance of 80.7% mIoU under full fine-tuning. In addition, its generalization and open-world perception potential are verified through video point cloud expansion and CLIP language space mapping.

**Questions:**

1. **Cross-modal generalization**: In the LiDAR-vision heterogeneous sensor fusion scenario, is an additional calibration step required?
2. **Long-term tracking performance**: How to solve the drift problem of the video extension version in long sequences?
3. **Lightweight potential**: Can a parameter compression version (such as distillation or quantization) be launched for mobile devices?

**Ethical Concerns:**

["NO or VERY MINOR ethics concerns only"]

**Final Justification:**

The authors' responses have addressed most of my concerns. Taking into account the overall evaluation of the other reviewers, I choose to maintain my rating as Borderline Accept (4).

**Limitations:**

1. **Scene constraints**: Relying on high-quality image-point cloud pairing data, not robust enough for textureless or low-light scenes.
2. **Semantic gap**: Language mapping only stays at linear projection, and does not model the complex semantic association between spatial concepts and natural language.
3. **Ecological dependence**: Highly dependent on 2D models such as DINOv2. If other backbones (such as Swin Transformer or SAM) are used, will the performance be stable?

**Paper Formatting Concerns:**

After review, no obvious format errors

**Quality:**

4

**Strengths And Weaknesses:**

**Advantages**:
1. Multimodal collaborative innovation: simulate human multi-sensory learning mechanism, break through the limitations of single-modal representation through cross-modal alignment (image-point cloud) and intra-modal distillation (point cloud feature optimization), and fill the gap between 2D and 3D features.
2. Comprehensive performance leadership: In tasks such as semantic segmentation and instance segmentation, Concerto significantly surpasses single-modal models such as DINOv2 and Sonata, and even outperforms feature splicing baselines, proving the effectiveness of multimodal collaboration.
3. Strong technical universality: supports pure point cloud input, video sequence expansion and language space mapping, adapts to multiple scenarios such as autonomous driving and robots, and has industrial landing potential.

**Disadvantages**:
1. Modality dependency limitations: cross-modal alignment depends on camera parameters, and performance may degrade in scenarios without calibration or multi-view missing, and the extreme case of pure point cloud without image input has not been verified.
2. High computational cost: 40k point clouds + 300k images are required for pre-training, and the video memory usage and training time are significantly higher than those of single-modal models. The ablation experiment did not quantify the difference in computational overhead.
3. Shallow language alignment: CLIP space is only mapped by linear projection, and the zero-shot segmentation accuracy (44.56% mIoU) is far behind the supervised results, and deep semantic fusion needs to be strengthened.

---

> ### Author Rebuttal · Authors · 2025-07-31
>
> Thank you for your kind acknowledgment of our work, especially for recognizing our model’s technical universality. We hope the responses provided address your concerns effectively.
>
> ### **W1: Modality Dependency Limitations**
>
> It is crucial to consider the potential lack of data from certain modalities in multi-modal joint learning. To address scenarios where images or calibration data may be missing, we ablate the image usage ratio of the whole datasets(ranging from 0\% to 100\%) in the pretraining stage in ***Table 5(b)*** of the paper. The results demonstrate Concerto's ability to adapt to point cloud data even in the absence of image input. Additionally, we include VGGT-lifted point clouds into our datasets, which makes it possible to directly use images to reconstruct without camera calibration. We provide more quantitative results of this variant:
>
> | Data Scale | ScanNet Val | ScanNet++ Val |
> | --- | :---: | :---: |
> | Concerto(lin.) | 77.3 | 45.7 |
> | +video training data(lin.) | **77.8** | **46.3** |
>
> We will include the results for this variant in the Appendix of the final version.
>
> Moreover, Concerto is robust to data with some degree of imperfection, such as noisy camera parameters, which is an advantage of our method. We begin with point fusion and then learning, which means we perform intra-modal self-distillation and cross-modal joint-embedding prediction at the abstract level, between fused points and image patches, instead of points and pixels. This approach ensures that slight misalignments between points and pixels do not cause significant differences in the loss calculations. Just as humans can understand noisy point clouds, Concerto shows a similar capability.
>
> ### **W2: High Computational Cost**
>
> Similar to DINOv2, Concerto is a pretrained self-supervised learning model that alleviates the heavy burden of downstream task learning. High computational cost at pretraining stage leads to strong and generalizable representations, resulting in data efficiency in downstream tasks. For example, ***Table 2(b)*** shows the strong linear probing results with less than 0.2M training parameters, ***Table 2(c)*** highlights outstanding performance in data-scarce situations, and ***Table 9 of the Appendix*** presents efficient LoRA-based adaptation.
>
> ### **W3: Shallow Language Alignment**
>
> Thank you for pointing that out. This language alignment is a future expectation for next-level evaluation criterion for self-supervised learning beyond single modality, which means although the model is trained without texts, we hope it can form concepts akin to human language.
>
> Since it is an evaluation protocol, we intentionally use a shallow language alignment with a linear layer to prevent other components, such as complex decoders, from dominating the final results. The detailed explanation behind this is discussed in the ***Pilot Study*** section of the paper.
>
> ### **Q1: Cross-Modal Generalization**
>
> Compared with the RGBD camera commonly used in indoor scenes, LiDAR + camera needs standard spatial-temporal synchronization, including same-time triggering (temporal) and intrinsic and extrinsic calibration(spatial). Our methods need no additional calibration.
>
> ### **Q2: Long-Term Tracking Performance**
>
> In our work, we use VGGT to lift videos from the RE10K dataset. The videos in RE10K are divided into short sequences, and VGGT effectively handles these sequences for point cloud generation. Although sometimes the feed-forward reconstruction produces failure cases, Concerto shows robustness when training on these noisy data. We also look forward to further improvements in feed-forward reconstruction methods and will release our code for generating this data.
>
> ### **Q3: Lightweight Potential**
>
> Thank you for your suggestion! We recognize the demand for lightweight models suitable for mobile devices. Currently, we have models of varying scales, and we plan to release these versions in the future. Below are the quantitative results for these models using linear probing:
>
> | Model Size | ScanNet Val | ScanNet200 Val | ScanNet++ Val | S3DIS Area5 |
> | --- | :---: | :---: | :---: | :---: |
> | 16M(T) | 67.7 | 25.0 | 33.7 | 65.2 |
> | 39M(S) | 76.6 | 34.4 | 43.1 | 71.3 |
> | 108M(B) | 77.3 | 37.4 | 45.7 | 73.5 |
> | 207M(L) | 77.5 | 38.6 | 46.3 | 73.7 |
>
> ### **L1 and L2:**
>
> Please refer to our response to W1 and W3 for detailed clarification.
>
> ### **L3: Ecological Dependence**
>
> We conducted experiments on different 2D encoders, like SigLIP2 [1] and RADIOv2.5 [2], as ***Table 8*** in the Appendix of our paper. The results show that the DINOv2 variant has the strongest performance, while the other encoders show comparable numerical performance.
>
> The choice of DINOv2 as the 2D encoder in our pipeline is based on its strong performance as a self-supervised model. We observed that different modal representations from self-supervised learning benefit the 3D intra-modal self-distillation significantly. This is analogous to the human cognition process: the learning for one single modality is like an intra-modal self-supervised learning process, while humans learn concepts through multi-modal inputs that interact in the system and result in superior recognition of the world. By integrating self-supervised features from two different modalities in Concerto, it emerges more reliable representations than single modal representations and their naive concatenation.
>
> [1] Tschannen, et al. SigLIP 2: Multilingual vision-language encoders with improved semantic understanding, localization, and dense features. arXiv:2502.14786 (2025).
>
> [2] Heinrich, et al. RADIOv2.5: Improved Baselines for Agglomerative Vision Foundation Models. CVPR 2025

---

> > ### Comment · Reviewer_yNfS · 2025-08-03
> >
> > Thanks to the author for your response, which has resolved most of my questions. Based on your response, I recommend you add the following related work:
> >
> > 1. Inter-modal masked autoencoder for self-supervised learning on point clouds, IEEE TMM, 2023
> > 2. Joint semantic segmentation using representations of LiDAR point clouds and camera images, INF FUS, 2024
> > 3. Self-supervised learning of lidar 3D point clouds via 2D-3D neural calibration, IEEE TPAMI, 2025

---

> > > ### Author Response · Authors · 2025-08-03
> > >
> > > Thank you for your acknowledgement of our response! We will add the related works you mentioned in our final version.

---

### Official Review · Reviewer_yoVm · 2025-06-25

**Clarity:** 2
**Significance:** 3
**Originality:** 4
**Rating:** 4
**Confidence:** 3

**Summary:**

This paper presents Concerto, a minimalist yet powerful framework for learning spatial representations by combining 3D intra-modal self-distillation with 2D-3D cross-modal joint embedding. I found the central idea of connecting 2D and 3D feature spaces particularly novel and compelling. The intuition that humans can abstract and recall spatial concepts across modalities is well-motivated and convincingly translated into the model design.

* The empirical results are somewhat strong: Concerto demonstrates fine improvements over both 2D and 3D SOTA baselines in fine-tuned settings
---> while the zero shot evaluation is missing and lacking.

Overall, the method is conceptually sound and has fine practical relevance.

That said, there are several comments that should be addressed prior to publication.

**Questions:**

Very important questons related also to the Weaknesses:


  1. Line 92–94 (Sentence Starting with "Ideally"): This sentence is confusing and seems to contradict the surrounding paragraph. Please clarify its intent and explain how it aligns with the rest of the section.

   2. Lines 112–113 – Clarification Needed: It took time to uderstand what is meant by "linear probing" in this context. you are projecting point cloud features into the CLIP feature space, given that CLIP already aligns images and text during its pretraining -- the goal is to align the 3d representation with text. The question is
 * why CLIP if you dont use the image embeddings? or do you?
 * can you use any other text feature extractor?
  * How to do align, what is the loss here?
 I think this pargraph need a better writing.

3. DINOV2 as a Target – Clarification: If your model is trained to predict DINOv2 features, does this imply that DINOv2 defines an upper bound for your performance? If so, how is your method able to outperform it? I found this somewhat confusing.

4. Lines 145–146 – Ambiguity: The phrasing “divide a scene into one point cloud” is unclear. Please revise this for clarity. Do you mean that each scene is represented as a single point cloud - then what does divide mean? If not, please specify the intended meaning.

**Ethical Concerns:**

["NO or VERY MINOR ethics concerns only"]

**Final Justification:**

From the beginning, I liked the paper, particularly the main idea, which I find both timely and important. I had some questions and concerns, but the authors addressed them satisfactorily. Therefore, I vote for acceptance.

**Limitations:**

See my Weaknesses 4 5 and 6.

**Quality:**

3

**Strengths And Weaknesses:**

** Strengths **
1. Novel and Well-Motivated Idea: The paper introduces a creative and compelling approach that bridges 2D and 3D modalities for spatial representation learning. The core intuition is both original and conceptually sound.

2. Good Experimental Results and Ablation: The experimental evaluation is comprehensive, with solid performance across multiple benchmarks. The ablation studies effectively isolate the contributions of different components (I appriciate this part very mach)*

3. Data Efficiency Results – A Strength: The results on data efficiency are excellent and, in my view, one of the paper’s key strengths. I suggest highlighting this more prominently in both the abstract and conclusion.

** Weaknesses **

1. Writing
  * Redundancy: The manuscript suffers from a noticeable degree of textual redundancy. I recommend a thorough revision to reduce repetition and improve clarity throughout the paper.

  * Lack of Formal Definitions: While the paper is generally well-written and conceptually clear, it would benefit from a more formal and mathematical presentation. Providing explicit formulas at each step of the pipeline would make the method easier to follow and reproduce.

* Some Statements Are Confusing: Certain parts of the paper are unclear or potentially contradictory. See Questions 1 and 2 in my review for specific examples that should be clarified.

2. Figure 3 – Needs Improvement: This figure is central to the paper, but currently it is unclear and potentially misleading. If it is meant to depict the architecture, it should be expanded with clearer annotations, supporting formulas, and step-by-step explanations. Consider reducing textual redundancy elsewhere in the paper to make room for a more informative and detailed illustration.

3. Intra-Modal Self-Distillation Section: This section is underexplained, despite being a key preliminaries of the method. Please provide more details on how intra-modal self-distillation is implemented and how it contributes to the learned representations.

4. Zero-Shot Segmentation – Untapped Potential: Have you considered evaluating your model on zero-shot segmentation tasks, similar to ConceptFusion [1] or FollowAnything [2], by providing an image or user click to specify the object of interest and not only text?

5. Long-Tail Concepts – Missing Evaluation: It would be informative to test the method on long-tailed or rare object classes, as found in datasets like unCOCO. This would demonstrate robustness in more realistic, open-world scenarios.

6. Comparison to ConceptFusion: ConceptFusion is a closely related work that also aligns 2D and 3D representations and achieves strong results on ScanNet and many other dtasets. A direct comparison would strengthen the paper, especially in light of similar goals. For zero short text, click, images as input.

[1] https://concept-fusion.github.io/
[2] https://arxiv.org/abs/2308.05737

---

> ### Author Rebuttal · Authors · 2025-07-31
>
> Thanks for your acknowledgement of our work, especially your recognition of our data-efficient and ablation results. And we hope your concerns are well addressed below.
>
> ### **W1: Clarity of Writing**
>
> - Redundancy
>
>     We will polish our paper to make it more concise and cohesive. We would also welcome more specific feedback.
>
> - Lack of Formal Definitions
>
>     Yes, some math formulas will help readers understand. Here we provide the mathematical presentation of our method:
>
>     As shown in Fig. 3 of the paper, we obtain point representations $s_x$ and image representations $s_y$ from the point encoder and image encoder respectively, as
>
>     $$s_x = \Psi_{\text{pcd}}(x),$$
>     $$s_y = \Phi_{\text{img}}(y).$$
>
>     With camera parameters $z$ as conditions, the correspondence between the points and pixels can be captured, which is further explained in Appendix A.2. First, the intrinsic matrix $K$ and extrinsic matrix $[R|t]$, derived from $z$, are used to project 3D point coordinates $p_x$ to the 2D pixel coordinates $p_y$:
>
>     $$p_y = K[R|t] p_x.$$
>
>     Then, for each image patch $j$, we compute the mean of the features of points falling within it using
>
>     $$\hat{s}\_{y,j} = \frac{1}{|\mathcal{N}\_j|} \sum_{i \in \mathcal{N}\_j} s\_{x,i},$$
>
>     where $\mathcal{N}\_j$ is the set of indices of points whose projected coordinates $p_y$ fall within the $j$-th image patch, and $|\mathcal{N}\_j|$ is the number of such points. $s\_{x,i}$ is the representation of the $i$-th point. Then, we compute the loss $D(s_y, \hat{s}_y)$ using cosine similarity
>
>     $$D(s_y, \hat{s}_y) = \left\langle \frac{s_y}{\||s_y\||_2}, \frac{\hat{s}_y}{\||\hat{s}_y\||_2} \right\rangle$$
>
>     between representations of point clouds and image patches. This process introduces 2D self-supervised features into 3D self-supervised learning and stimulates the point intra-modal self-distillation process.
>
> - Confusing Statement
>
>     Please refer to our responses to Q2 and Q3 for clarification.
>
> ### **W2: Figure 3 Improvement**
>
> We will improve Figure 3 to present our method more clearly. As mentioned in W1, we will include a more detailed version of the formulas to complement the figure.
>
> ### **W3: Further Explanation for Intra-Modal Self-Distillation Section**
>
> We agree that introducing the preliminary work is important for readers, especially those unfamiliar with it. Here’s a more detailed explanation:
>
> In Intra-Modal Self-Distillation, we build upon the principles of multi-view consistency and masked reconstruction, illustrated in Part (a) of Figure 3. The main challenge in 3D point cloud self-supervised learning is the geometry shortcut, which means networks tend to focus on sparse point cloud geometry and neglect the rich semantic information, such as color and texture. To mitigate this, it is important to make the geometry clues harder for the network to capture. This approach is discussed in detail in the Sonata paper.  In addition to applying masking, cropping, and view rolling techniques, we also adopt the encoder-only structure and feature upcasting strategy from Sonata to reduce the over-reliance on local geometry and incorporate multi-scale features. This preliminary step ensures the point encoder refines its representations with richer semantic information.
>
> We will revise this section, provide a detailed explanation in the Appendix, and refer readers to the Sonata [1] paper for more information.
>
> ### **W4: Zero-Shot Segmentation**
>
> Thank you for the suggestion to further improve our work. ConceptFusion [2] and FollowAnything [3] are both remarkable contributions to zero-shot semantic segmentation using existing foundation models(e.g., SAM, DINO, and CLIP) and providing various interfaces for users. However, we focus on building such a foundation model in the 3D domain, like DINOv2 in the 2D domain. The absence of such a 3D foundation model blocks the progress in improving downstream tasks. We acknowledge this as a key area for future work and will continue to push in this direction.
>
> Additionally, we have noticed that evaluation datasets like unCoCo, used in ConceptFusion and FollowAnything, are currently not available. However, we provide a comparison between our method and ConceptFusion on ScanNet, as mentioned in W6.
>
> ### **W5: Long-Tail Concepts**
>
> We agree that long-tail concepts are crucial for evaluating the generalization ability of model. Although the unCoCo dataset is currently unavailable, we believe that the results on ScanNet200 in our paper can offer insights into the model's performance on long-tail concepts. ScanNet200 includes more rare classes than ScanNet, and our model achieves 37.4\% mIoU on ScanNet200 using linear probing, compared to the previous SOTA of 29.3\%. This improvement demonstrates Concerto's effectiveness in handling long-tail objects.
>
> ### **W6: Comparison to ConceptFusion**
>
> We followed the evaluation protocol from ConceptFusion, using the same subset and object classes without background concepts. Below is a comparison between Concerto and ConceptFusion on ScanNet Val:
>
> | Methods | mAcc | mIoU |
> | --- | :---: | :---: |
> | ConceptFusion | 0.63 | 0.58 |
> | Concerto | 0.75 | 0.58 |
>
> However, the main focus of Concerto is point cloud representation learning. The language probing serves as an auxiliary evaluation method to demonstrate the potential of our representations. This shows that even without explicit text information, the representations learned by our method can inherently capture conceptual features related to language. With simple alignment, these representations can be transformed into text space. We intentionally use shallow language alignment to avoid complex components, such as decoders, dominating the evaluation results.
>
> For W4-W6, we will include the related results and discussion in our final version.
>
> ### **Q1: Lines 92–94 (Sentence Starting with "Ideally")**
>
> This sentence refers back to Figure 2, which illustrates the "Apple" concept in cognition. Humans are exposed to multi-modal information from birth, and once they learn the concept of an apple, they can recall its taste when they see a picture of it. We aim to express a similar idea here. Specifically, while the model is trained using multi-modal joint representation learning, it is capable of outputting representations derived from 2D and 3D information, even when the input is a single modality, such as a 3D point cloud, during inference. This means a lot in real-world applications, because multi-modal input is not always available for a system.
>
> We will rewrite this part in the final version to erase the confusion.
>
> ### **Q2: Lines 112–113 – Clarification**
>
> This paragraph describes our future expectation for a next-level evaluation criterion for self-supervised learning beyond single modality, which means that although the model is trained without texts, it can simultaneously form concepts akin to human language. Just as humans first learn concepts from the world before using language to express them, a strong self-supervised model should be able to form its own conceptual "language". This property can be validated during evaluation on text alignment, where the model’s representations can be aligned with the human semantic space through a simple transformation, such as a single linear layer. As mentioned in the W6, aligning representations with text is not the main contribution of this work but rather an auxiliary evaluation method to validate the quality of Concerto's representations, complementing the semantic segmentation tasks.
>
> Then, come to the question. We here choose the CLIP text space to validate our representations. We align our point cloud representations with the image embeddings from LSeg, using the image as a bridge to connect our 3D point features with text embeddings. If any other text feature extractor is aligned with image pixels, Concerto’s representations can be transformed into that text space as well. The loss function used for alignment is cosine similarity. We freeze the pretrained backbone, add a linear layer on top, and align the output with the image embeddings from the LSeg image encoder.
>
> ### **Q3: DINOV2 as a Target – Clarification**
>
> Our model does not aim to simply copy 2D features from DINOv2, which ignores 3D information. Instead, we emphasize the importance of interaction between modalities. This leads to entirely new representations derived from different domains. The core of our method is the stimulation from the 2D modality to enhance intra-modal self-distillation in the 3D domain. As shown in ***Table 1*** of the paper, our method outperforms each single-modality approach as well as their naive concatenation. This demonstrates that, rather than merely combining information from two modalities, Concerto captures richer representations through the interaction between 2D and 3D information.
>
> ### **Q4: Lines 145–146 – Ambiguity**
>
> Apologies for the ambiguity. What we meant to convey is that a scene with multiple images (>>4) is divided into several data pieces, where each piece consists of one point cloud and 4 images. We will revise this section in the final version to ensure better clarity.
>
> [1] Wu, et al. Sonata: Self-Supervised Learning of Reliable Point Representations. CVPR 2025.
>
> [2] Jatavallabhula, et al. ConceptFusion: Open-set multimodal 3d mapping. RSS 2023
>
> [3] Maalouf, et al. Follow Anything: Open-set detection, tracking, and following in real-time RA-L 2024

---

### Official Review · Reviewer_NTix · 2025-06-28

**Clarity:** 3
**Significance:** 2
**Originality:** 2
**Rating:** 4
**Confidence:** 4

**Summary:**

This paper presents Concerto, a framework motivated from multi-sensory learning in humans, that aims at achieving better spatial representations for self-supervised point cloud features. It first self-distills the 3D model through a teacher-student paradigm, then utilizes 2D-3D joint embedding prediction to align the 3D point cloud features with the 2D image features. Experimental results show the superiority of the proposed Concerto when applying to downstream tasks of semantic and instance segmentation under different tuning settings including linear probing, decoder probing, and full finetuning.

**Questions:**

-- How is the proposed method compared with other methods that also focus on spatial understanding with joint 2D-3D data like [1, 2, 3, 4]? The comparisons are important to understand which is a better solution for joint 2D-3D spatial understanding.

-- Is the pretrained 2D model also tunable during training? If not, I think it cannot be called "synergy emerged from joint self-supervised learning". If so, will the 2D model gets improved from jointly training with the 3D model?

-- Does there exist difference from the self-distillation stage presented in the paper and the self-distillation operation in the Sonata paper? If not, I think this stage cannot be an essential component of the proposed method, because the proposed method is built upon Sonata, while the self-distillation stage is just a part of the original Sonata paper.

-- For the Concerto variant specifically for video, is the only difference of this model and the original model being the training data is constructed from video using VGGT instead of point cloud? If so, why this is treated as a separate variant, instead of adding the video data to the training dataset to enrich the training set, and get a more powerful model with the more diverse training set?

[1] Fan et al. Large Spatial Model: End-to-end Unposed Images to Semantic 3D. NeurIPS 2024.

[2] Yue et al. Improving 2D Feature Representations by 3D-Aware Fine-Tuning. ECCV 2024.

[3] Zhu et al. PonderV2: Pave the Way for 3D Foundation Model with A Universal Pre-training Paradigm. arXiv:2310.08586.

[4] Wang et al. Open Vocabulary 3D Scene Understanding via Geometry Guided Self-Distillation. ECCV 2024.

**Ethical Concerns:**

["NO or VERY MINOR ethics concerns only"]

**Final Justification:**

My major concern on the performance compared with other spatial understanding methods is well resolved in the rebuttal. The remaining issues are about the contribution and motivation. For paper contribution, since the authors say that the "intra-modal self-distillation" part is not a newly proposed component in this paper (but it is included in the method, and make readers feel that it is a new technique), the only technical contribution of this paper is the "2D-3D joint embedding prediction", which makes the paper contribution a bit limited. Also, the motivation from human perception for 2D-3D joint modeling is too far-fetched and very loosely related. But anyway, the simplicity and effectiveness of the proposed solution make the strengths of this paper outweigh the weaknesses. Therefore, I would like to raise my score to borderline accept.

**Limitations:**

Yes

**Quality:**

2

**Strengths And Weaknesses:**

**Strengths**:

++ The idea of using Joint Embedding Predictive Architecture (JEPA) to predict 3D features from pretrained 2D encoders is neat and effective solution to me.

++ The experimental settings are comprehensive, including different tuning strategies of linear probing, decoder probing, and full finetuning, demonstrating the broad use case of the proposed Concerto under different circumstances and training budgets. Also, the model almost always outperforms the baseline Sonata, showing superior performance on downstream tasks.

++ The ablation study on multiple aspects including criteria type and weight, the ratio of point clouds and images, *etc.* of the implementation can be useful for guiding towards the optimal configuration of the model.

**Weaknesses:**

-- The paper lacks the comparisons with other methods that also focus on spatial understanding with joint 2D-3D data, essentially the methods listed in the third paragraph of the related work section. More specifically, Large Spatial Model [1], FiT3D [2], PonderV2 [3], GGSD [4] are some open-source options, and also representing the different categories listed in the related work section. If the original works are on 2D segmentation, we can lift the 2D features to 3D using camera poses for comparison in 3D segmentation. This comparison is very important because they are actually the existing works for spatial understanding with joint 2D-3D data, while the current compared methods like MSC, Sonata are all pure 3D self-supervised methods. It will help demonstrate how the proposed solution for joint 2D-3D learning performs compared to alternative solutions in other works like lifting projections, differentiable rendering, direct distillation, *etc.*

-- The paper claims to have synergy between the 2D and 3D models. However, from my understanding of the method, the 2D model is fixed, so the model is essentially getting knowledge from pretrained 2D model to predict a better 3D model, but there is no benefit the other way around. Therefore, I think the synergy from the joint learning paradigm is overclaiming.

-- The 3D intra-model self-distillation part follows the identical approach of previous work Sonata. Therefore, I think it cannot be regarded as part of the proposed method. However, the paper gives a feeling that the proposed method has two steps "(a) intra-modal self-distillation + (b) 2D-3D joint embedding prediction". In fact, only (b) is the technique proposed in the paper.

-- It sounds a bit far-fetched to me to connect the proposed method with the human concept learning in cognition. Basically the paper is proposing a paradigm for 2D-3D joint learning, which is just very loosely related to the multi-sensory learning concept of humans illustrated in the paper. It is definitely good to have some bio-inspired motivations or designs in the method, but I think the concept is very loosely related to the proposed method. Otherwise, any method that introduce additional modality in its framework can be related to human multi-sensory learning.

-- In the supplementary the paper mentioned the term "DINOv2.5". However, I think this is not a standard term. I could only refer to the Sonata paper where they refer to [5] as DINOv2.5. I think a more standard term for this work is called DINOv2-Reg. Also, the submitted manuscript does not have reference about [5], which makes readers feel confusing.

[1] Fan et al. Large Spatial Model: End-to-end Unposed Images to Semantic 3D. NeurIPS 2024.

[2] Yue et al. Improving 2D Feature Representations by 3D-Aware Fine-Tuning. ECCV 2024.

[3] Zhu et al. PonderV2: Pave the Way for 3D Foundation Model with A Universal Pre-training Paradigm. arXiv:2310.08586.

[4] Wang et al. Open Vocabulary 3D Scene Understanding via Geometry Guided Self-Distillation. ECCV 2024.

[5] Darcet et al. Vision Transformers Need Registers. ICLR 2024.

---

> ### Author Rebuttal · Authors · 2025-07-31
>
> Thanks for your suggestions. We are sincerely grateful for your efforts and feedback. We hope the concerns are well addressed.
>
> ### **W1: Comparisons with Other Methods on Spatial Understanding with Joint 2D-3D Data**
>
> Thanks for your suggestion. In response, we’ve included comparisons between Concerto and the methods you mentioned on spatial representation learning:
>
> | Methods             | ScanNet Val | ScanNet200 Val | ScanNet++ Val | S3DIS Area5 |
> | :------------------ | :---------: | :------------: | :-----------: | :---------: |
> | Large Spatial Model |    73.6     |      28.4      |     42.0      |    70.0     |
> | Ponderv2            |    77.0     |      32.3      |     46.0      |    73.2     |
> | FiT3D               |    44.8     |      24.1      |     32.8      |      -      |
> | Concerto            |  **80.7**   |    **39.2**    |   **50.7**    |  **77.4**   |
>
> Large Spatial Model [1], PonderV2 [2], and FiT3D [3] represent different categories of joint 2D-3D learning, such as lifting projections, differentiable rendering, and direct distillation. For GGSD [4], since it focuses on open-vocabulary tasks using additional text and superpoint information, which is outside the scope of our current work, it is not included in this comparison. We will continue to extend our work to other downstream tasks in the future.
>
> ### **W2: Synergy from the 2D and 3D**
>
> Thank you for your feedback on our use of the term "synergy". To address your concern, we provide results on ScanNet where DINOv2 is fine-tuned with Concerto, showing an improvement in 2D tasks compared to the original DINOv2. This demonstrates that fine-tuning the 2D model alongside the 3D model enhances its performance on 2D tasks.
>
> | 2D | mIoU | mAcc | allAcc |
> | --- | :---: | :---: | :---: |
> | DINOv2 | 63.1 | 75.5 | 82.4 |
> | DINOv2(Concerto) | **64.5** | **77.4** | **83.2** |
>
>
> **Here, when we refer to "synergy," we mean "1 + 1 > 2"**: with joint learning, our method allows for stronger and more reliable representations than what can be achieved with single modalities or the naive concatenation of 2D and 3D features. This comparison can be seen in ***Table 1*** of our paper.
>
> We acknowledge that our initial expression may have caused some misunderstanding about the nature of our method. We will revise the relevant section in the final version to make this clearer and will include the fine-tuning results for DINOv2 in the Appendix.
>
> ### **W3: 3D Intra-Model Self-Distillation as Preliminary**
>
> Thanks for your suggestion. The Intra-Modal Self-Distillation part is not a new contribution of this work, but rather a preliminary step. We introduced it here for clarity and to help readers better understand the foundational aspects of our method. A preliminary introduction is necessary for those who are not familiar with previous works. The other Reviewer, yoVm, also thinks this preliminary should be explained in more detail. In the final version, we will explicitly clarify that the intra-modal self-distillation is a preparatory step and not the primary contribution of our work. Our main contribution lies in the Cross-Modal Joint Embedding Prediction and its integration with self-distillation.
>
> ### **W4: Human Concept Learning in Cognition**
>
> Thank you for your advice. We will revise the explanation of human concept learning to improve clarity and the overall flow of the manuscript.
>
> We introduce the concept of human cognition to help readers better understand the awareness in self-supervised learning with the other modality. Humans are exposed to multi-sensory information from birth, and once they learn a concept, such as an "apple," they can recall its taste when they see the fruit. In real-world applications, however, multi-sensory information is not often available to a system. This raises the question: How can we generate multi-sensory representations from a single modality, much like how humans recall information from different senses in their mind?
>
> Our goal is to present a model that can produce representations derived from both 2D and 3D features with only point cloud input, mimicking the process by which humans recall information from other sensory modalities. This is different from the case where one model receives multi-modal input and outputs corresponding responses.
>
> ### **W5: Terms for DINOv2-Reg**
>
> Thank you for pointing this out. We will update the term DINOv2-Reg [5] in the final version to ensure consistency and clarity, and include the reference.
>
> ### **Q1-3:**
>
> Thank you for your questions. For detailed responses, please refer to the corresponding sections addressed in the previous  W1-3.
>
> ### **Q4: Concerto Variant Specifically for Videos**
>
> We appreciate your suggestions. To address the data limitations hindering progress in 3D self-supervised learning, we explore the use of abundant online video data. By employing feed-forward methods like VGGT, we can enlarge our point cloud datasets. To demonstrate the potential of this approach, we introduce a separate variant of Concerto that incorporates this additional video data.
>
> For quantitative results of VGGT-lifted video data, we provide the results using the base model on ScanNet and ScanNet++:
>
> | Data Scale | ScanNet Val | ScanNet++ Val |
> | --- | :---: | :---: |
> | Concerto(lin.) | 77.3 | 45.7 |
> | +video training data(lin.) | **77.8** | **46.3** |
>
> Below, we provide mIoU evaluation results on ScanNet Val for different data scales using linear and decoder probing, showing scaling performance on video data. Here, 87k refers to 40k original point clouds and 47k VGGT-lifted point clouds.
>
> | Data Scale | 13k | 23k | 35k | 40k | 87k |
> | --- | --- | --- | --- | --- | --- |
> | lin. | 76.5 | 76.6 | 77.4 | 77.3 | 77.8 |
> | dec. | 78.5 | 78.9 | 79.3 | 79.5 | 79.8 |
>
> The table above shows that, with additional video data, Concerto can further improve the performance. We believe that with the progress of feed-forward methods, the point cloud self-supervised learning has the potential to scale up to a larger level and finally result in a more reliable and robust spatial representation model. We will append these results to the Appendix in the final version.
>
> [1] Fan et al. Large Spatial Model: End-to-end Unposed Images to Semantic 3D. NeurIPS 2024.
>
> [2] Zhu et al. PonderV2: Pave the Way for 3D Foundation Model with A Universal Pre-training Paradigm. T-PAMI 2025
>
> [3] Yue et al. Improving 2D Feature Representations by 3D-Aware Fine-Tuning. ECCV 2024.
>
> [4] Wang et al. Open Vocabulary 3D Scene Understanding via Geometry Guided Self-Distillation. ECCV 2024.
>
> [5] Darcet et al. Vision Transformers Need Registers. ICLR 2024.

---

> > ### Comment · Reviewer_NTix · 2025-08-03
> >
> > Thanks the authors for the rebuttal! My major concerns regarding the model performance (e.g., comparison with other methods) are solved. For "3D Intra-Model Self-Distillation as Preliminary", since it is the case, I feel the contribution of this paper becomes a bit limited as then "2D-3D joint embedding prediction" is the only module proposed in the paper. Also, for connecting the proposed method with human perception, I still do not see a tight relationship, because otherwise any method that perform multi-modal learning (or containing 2D&3D input modalities) can call themselves "related to human perception".
> >
> > However, I do honor the merit of this paper having a straightforward and effective solution that achieves superior performance. With that being said, I would like to raise my rating to 4.

---

> > > ### Author Response · Authors · 2025-08-04
> > >
> > > Thank you for acknowledging our rebuttal to your major concerns and willingness to change the score to 4. We will revise the related part according to your review in our final version.

---

### Official Review · Reviewer_T3Xs · 2025-07-01

**Clarity:** 2
**Significance:** 3
**Originality:** 3
**Rating:** 5
**Confidence:** 3

**Summary:**

This paper proposes Concerto, which extends a 3D self-supervised learning (SSL) method (Sonata) by cotraining with 2D image features via joint embedding predictive objective, and demonstrate consistent performance gain on various tasks, surpassing the single modality performance and naive concetenation of 2D and 3D features, providing a straightforward and effective way of enhancing 3D SSL encoders.

**Questions:**

1. Does batch size matter in the cross-modal training?
2. Can the method work with noisy camera parameters? Will it be sensitive if depth, camera poses or intrinsics are inaccurate?

**Ethical Concerns:**

["NO or VERY MINOR ethics concerns only"]

**Final Justification:**

Concerns are addressed. Decide to stay at the original rating "accept". Detailed justification has been replied to the authors.

**Limitations:**

yes

**Paper Formatting Concerns:**

No formatting concerns noticed.

**Quality:**

4

**Strengths And Weaknesses:**

**Strengths:**
1. Straightforward and effective method. SSL on 2D image features has witnessed a substantial progress. Combing their power to improve 3D features has sigfinicance to the community.
2. Consistent performance gain. Concerto demonstrates consistent performance gain over baselines on various benchmarks, providing a strong empirical evidence.
3. Ablation studies covered many aspects.

**Weaknesses:**
1. Clarity of writing can be improved if more important details can be included in the main paper. Specifically:
 - 1a) how are the image and point feature represented when computing JEPA? If would be nice to have some concise and clear math expressions and formulations.
- 1b) In ablation study, what is the definition of "image usage"?
- 1c) In ablation study, how do you specifically control the number of visible points and how is the specific number 65536=256^2 chosen? How do you get the number from the visibility check in the appendix?
- 1d) Lack of implementation details on the learning probing and decoder probing.
2. Performance seems to saturate when scaling up data and model, in Table 5h. Scaling is an important indicator but only two settings are ablated. It would be great if the authors can provide more results on this, perferablly a plot and some explanations.
3. Lack of quantitative results of training on the VGGT lifted video data. It would be great to see the method can robustly leverage VGGT lifted data as training source and possibly showing some scale performance on the video data where data is abundant while annotation is scarce.

---

> ### Author Rebuttal · Authors · 2025-07-31
>
> Thanks for your acknowledgment of our work! Below, we provide detailed responses to your questions and comments. Hopefully, they will further clarify our approach.
> ### **W1: Clarity of Writing**
>
> - Clarity of Writing
>
>   Thanks for your suggestion. We agree that adding math formulas will make the process clearer for readers, and we will revise this section in the final version.
>
>   As shown in Fig. 3 of the paper, we respectively obtain point representations $s_x$ and image representations $s_y$ from the point encoder and image encoder, as
>
>   $$s_x = \Psi_{\text{pcd}}(x),$$
>   $$s_y = \Phi_{\text{img}}(y).$$
>
>   With camera parameters $z$ as conditions, the correspondence between the points and pixels can be captured, which is further explained in Appendix A.2. First, the intrinsic matrix $K$ and extrinsic matrix $[R|t]$, derived from $z$, are used to project 3D point coordinates $p_x$ to the 2D pixel coordinates $p_y$:
>
>   $$p_y = K[R|t] p_x.$$
>
>   Then, for each image patch $j$, we compute the mean of the features of points falling within it, as
>
>   $$\hat{s}\_{y,j} = \frac{1}{|\mathcal{N}\_j|} \sum_{i \in \mathcal{N}\_j} s_{x,i},$$
>
>   where $\mathcal{N}\_j$ is the set of indices of points whose projected coordinates $p_y$ fall within the $j$-th image patch, and $|\mathcal{N}\_j|$ is the number of such points. $s\_{x,i}$ is the representation of the $i$-th point. Then, we compute the loss $D(s_y, \hat{s}_y)$ using cosine similarity
>
>   $$D(s_y, \hat{s}_y) = \left\langle \frac{s_y}{\||s_y\||_2}, \frac{\hat{s}_y}{\||\hat{s}_y\||_2} \right\rangle$$
>
>   between representations of point clouds and image patches. This process introduces 2D self-supervised features into 3D self-supervised learning and stimulates the point intra-modal self-distillation process.
>
> - Definition of "Image Usage"
>
>   The term "Image Usage" refers to the ratio of the total number of images used during the pretraining process. If the Image Usage is 50\%, it means we randomly selected 50\% of the data with images, while the remaining is pure point cloud data. We will clarify this definition in the final version of our paper.
>
> - Visible Points
>
>   The control of visible points is crucial in the self-supervised learning process. The model needs to capture local cues to predict surrounding representations. Too many visible points can diminish this exploration: as training progresses, random views of the same data are similar, which reduces diversity in the model’s learning. Conversely, too few visible points make the task unmanageable. The value of 65536 is chosen experimentally by visualizing randomly selected views from the point clouds.
>
>   Fully controlling the visible points for each data instance is impractical, as points in the point cloud do not have a one-to-one correspondence with pixels in the image. The chosen number of visible points (65536) is the maximum that ensures the model focuses on different regions of the data at an appropriate scale. Typically, the number of visible points falls within a user-defined proportional range of the total point count. Any count exceeding 65536 is capped at this maximum value.
>
>   We will update the related section in the final version for better clarity.
>
> - Linear Probing and Decoder Probing
>
>   **Linear Probing**: In this approach, the pretrained encoder remains frozen. We upcast the features to their original resolution and adapt them to downstream tasks using a single linear layer, which accounts for less than 0.2\% of the model’s total parameters.
>
>   **Decoder Probing**: For this method, we use a lightweight, hierarchical decoder that comprises 13\% of the total parameters. The encoder is frozen, and we train only the decoder, allowing it to learn task-specific representations.
>
>   We will add this explanation to the final version. As these methods are described in detail in the Sonata paper, we will also refer readers to the original Sonata [1] work for further clarification.
>
> ### **W2&3: Data Scaling and Video-Lifted Data Potential**
>
> Thanks for your suggestion. Currently, 3D self-supervised learning faces the challenge of limited data, which restricts further development. To address this, we explore leveraging large-scale video data through feed-forward methods like VGGT to expand the dataset. Below, we provide evaluation results (mIoU) on ScanNet Val using linear and decoder probing for different data scales:
>
> | Data Scale | 13k | 23k | 35k | 40k | 87k |
> | --- | --- | --- | --- | --- | --- |
> | lin. | 76.5 | 76.6 | 77.4 | 77.3 | 77.8 |
> | dec. | 78.5 | 78.9 | 79.3 | 79.5 | 79.8 |
>
>
> Here, 87k means 40k original point clouds and 47k VGGT-lifted point clouds. For quantitative results of VGGT-lifted video data, we provide the mIoU results with base model size on ScanNet and ScanNet++:
>
> | Data Scale | ScanNet Val | ScanNet++ Val |
> | --- | :---: | :---: |
> | Concerto(lin.) | 77.3 | 45.7 |
> | +video training data(lin.) | **77.8** | **46.3** |
>
> The table above shows that, with additional video data, Concerto can further improve the performance. As for model size, it should be increased with the data scale. Currently, the model is still in the stage of being data hungry. Therefore, we don't scale up our model size to a giant level but provide a tiny version for robotic downstream tasks and a large model trained with additional video data here. All the models are evaluated using linear probing with mIoU metric.
>
> | Model Size | ScanNet Val | ScanNet200 Val | ScanNet++ Val | S3DIS Area5 |
> | --- | :---: | :---: | :---: | :---: |
> | 16M(T) | 67.7 | 25.0 | 33.7 | 65.2 |
> | 39M(S) | 76.6 | 34.4 | 43.1 | 71.3 |
> | 108M(B) | 77.3 | 37.4 | 45.7 | 73.5 |
> | 207M(L) | 77.5 | 38.6 | 46.3 | 73.7 |
>
>
> We believe that, as feed-forward methods continue to improve, point cloud training will scale more effectively, leading to more robust and reliable spatial representations. We will append these scaling results in the appendix of the final version and release a HuggingFace demo and website with more visualizations showcasing performance on VGGT-lifted point clouds.
>
> ### **Q1: Does batch size matter in the cross-modal training?**
>
> While batch size is an important parameter in self-supervised learning, Concerto shows strong adaptability to different batch sizes during pretraining. The following table presents results across various batch sizes on different benchmarks:
>
> | Batch Size | ScanNet Val | ScanNet200 Val | ScanNet++ Val | S3DIS Area5 |
> | --- | :---: | :---: | :---: | :---: |
> | 48 | 78.0 | 38.2 | 46.9 | 73.9 |
> | 96 | 77.3 | 37.4 | 45.7 | 73.5 |
> | 192 | 77.9 | 36.4 | 45.3 | 73.8 |
>
>
> ### **Q2: Can the method work with noisy camera parameters? Will it be sensitive if depth, camera poses, or intrinsics are inaccurate?**
>
> Yes, our method is robust to noisy camera parameters, which is one of its advantages. In our approach, we do fusion first and then learning. This allows minor misalignments between points and pixels without significantly impacting the loss calculations. Just as humans can understand noisy point clouds, Concerto shows a similar capability. Current feed-forward reconstruction methods may not always produce accurate point clouds and camera parameters, but Concerto shows strong adaptability to such noisy point clouds. Notably, we avoid cherry-picking video examples in our ongoing demos and paper, further proving the robustness.
>
> [1] Wu et al. Sonata: Self-Supervised Learning of Reliable Point Representations. CVPR 2025.

---

> > ### Comment · Reviewer_T3Xs · 2025-08-05
> >
> > Thank you for the rebuttal. The authors rebuttal have largely addressed my concerns and questions. I have also browsed other reviews. Overall, I view this submission as a solid, though arguably a small and straightforward, techinical progress in the subfield. Therefore, I am willing to maintain my score as accept.

---

> > > ### Author Response · Authors · 2025-08-05
> > >
> > > Thank you for acknowledging our work and rebuttal! We will continue to refine our work to a better version according to your feedback.

---

### Note · Authors · 2025-08-14

Dear Reviewers and ACs,

Thank you for giving all positive feedback after the rebuttal and insightful discussion. Aiming to build a trustworthy foundational model for point cloud processing, we present Concerto.

During the review phase, we received encouraging feedback from the reviewers, particularly regarding:
- **Effective and Intuitive Method on 2D-3D Synergy**(All Reviewers): Concerto simulates human multi-sensory learning, using a JEPA-inspired architecture to achieve a 2D-3D synergy that surpasses single modalities or their naive fusion.

- **SOTA Performance and Detailed Ablations**(All Reviewers): Concerto achieves SOTA results across multiple benchmarks under various evaluation protocols (linear/decoder probing, fine-tuning), supported by extensive ablation studies.

- **Data and Parameter Efficiency**(Reviewer yoVm): Concerto excels in data-scarce scenarios, with LoRA-based adaptation and linear probing (<0.2M parameters).

- **Broad Applicability**(Reviewer yNfS): Concerto supports point clouds, video sequences, and language mapping, making it suitable for applications like robotics and navigation.

Based on your constructive feedback, we have worked together to improve our manuscript in the following aspects. We truly appreciate your engagement:
- **Quantitative Results for Scaling Up and Lightweight Potential**: We added quantitative results for scaling up with video data and new lightweight versions to show the model's potential directly.
- **Comparison with Other 2D-3D Methods**: We included comparison results across 4 benchmarks, proving the advancement of our methods.
- **Clarification of Synergy from Joint Learning**: We clarified that we use "synergy" to mean "1+1>2" (joint learning > sum of parts) and provided the results indicating that both 2D and 3D encoders improve during joint learning.
- **Zero-Shot Comparison with ConceptFusion**: We deliberately chose a simple linear layer as an evaluation metric and attached the comparison results between Concerto and ConceptFusion.
- **Explanation for Robustness to Imperfect and Uncalibrated Data**: Concerto tolerates noisy, uncalibrated data by learning from lifted video and fusing points before learning.

Finally, we reaffirm our commitment to conducting responsible research. We will release all code and pre-trained models to ensure full reproducibility and remain dedicated to contributing to its growth through continuous foundational research efforts.


Best,

Concerto Authors

---

### Decision · Program_Chairs · 2025-09-17

**Decision:**

Accept (poster)

**Comment:**

This paper presents a joint 3D self-supervised learning approach combined with 2D joint embedding predictive learning, demonstrating consistent improvements over alternative approaches across multiple tasks. The reviewers unanimously agreed on the value of the proposed method and the rigour of the presented evidence. The extensive comparisons with other models provide strong support for the added value of Concerto.

However, several presentation issues should be addressed in the camera-ready version. I agree with reviewer NTix that the connection with human perception is tenuous and unclear, and I recommend substantially downplaying or removing these claims in the revised paper. Additionally, as correctly noted by several reviewers, the text would benefit from revision to eliminate repetitive and redundant sections that detract from the clarity of the paper.

Despite these presentation concerns, the technical contributions are solid and the experimental validation is thorough. Given the positive evaluations from all reviewers and with the expectation that the camera-ready paper will address the identified presentation issues, I recommend acceptance.